# INTERLEAVING OPTIMIZERS FOR DNN TRAINING

## ABSTRACT

Optimizers are crucial in deep neural network (DNN) training, affecting model quality and convergence. Researchers have found that different optimizers often suit different problems or different stages of a problem. Hence, some studies have tried to combine different optimizers to better train DNNs. However, existing methods are limited to simple optimizer switch strategies, which leads to unstable model quality and slow convergence. In this paper, we propose a fine-grain optimizer switch method called Interleaving Optimizer for Model Training (IOMT), which automatically switches to the appropriate optimizer among different optimizer types based on the training stage information, achieving faster convergence and higher test accuracy. IOMT employs surrogate models to estimate the performance of different optimizers during training and is supported by a transferability assessment to predict the training cost. By combining the transferability assessment, performance estimation, and training process information with an acquisition function, IOMT calculates the optimization gain of each optimizer and switches the optimizer with the largest gain for the next training stage. The experimental results on full training and fine-tuning demonstrate that IOMT achieves faster convergence (e.g., 10% on the *stl10* dataset) and better performance (e.g., 3% accuracy improvement on the *cifar10* dataset) compared to existing methods.

## 1 INTRODUCTION

The choice of optimizer and its hyperparameter settings (e.g., the learning rate) profoundly impacts the model quality and convergence speed in deep neural networks (DNNs) (Soydaner, 2020; Hassan et al., 2023). Researchers typically use a single optimizer for the entire training (i.e., a coarse-grain optimizer setting) and have some empirical preferences for optimizer selection, such as using SGD for head fine-tuning (Poojary & Pai, 2019) and Adam for LoRA (Hu et al., 2021). However, recent studies find that different optimizers are not only suited to specific tasks but also exhibit unique characteristics and optimization strategies at different stages of a training (Im et al., 2016). Figure 1 presents the optimization results of three optimizers with varying runs (i.e., 200 times with different random seeds and hyperparameter settings) on four deterministic functions (rosenbrock, himmelblau, griewank and ackley). Different optimizers follow distinct paths in the same start point even with varying runs, making it difficult to definitively identify the "one size fits all" optimizer.

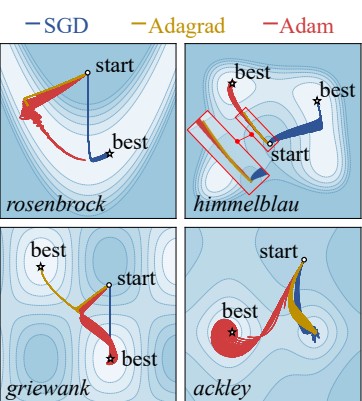

Figure 1: The different training processes with various optimizers.

To address such challenges of coarse-grain optimizer tuning, some studies have attempted to combine the benefits of different optimizers during a single training process recently. SWATS (Keskar & Socher, 2017) achieved better generalization by switching from Adam to SGD. Chen et al. proposed a partially adaptive momentum estimation method, which unifies the adaptive gradient methods (i.e., Adam or Amsgrad) with SGD by introducing a partial adaptive parameter (Chen et al., 2018). AdaBound (Luo et al., 2019) employed dynamic bounds on learning rates to achieve a gradual and smooth transition from adaptive methods to SGD. However, these approaches remain limited in the optimizer types (i.e., only two kinds of optimizers) and combining methods (i.e., simple switch strategy), which leads to unstable model quality and high training cost (Sun, 2020).

Based on the idea that "different optimizers suit for different parameter states", we propose a fine-grain optimizer switch method called **I**nterleaving **O**ptimizer for **M**odel **T**raining (IOMT). During the training, IOMT constructs surrogate models for different optimizers to predict their optimization benefits under various model parameter states. To better assess the benefits of the optimizers (i.e., potential loss reduction and convergence speed), IOMT calculates an optimization gain score for each optimizer using the acquisition function that combines the predicted performance, a transferability assessment, and training process information. By carefully switching the optimizer with the highest score during training, IOMT achieves faster convergence and better model quality. To summarize, the key contributions of this paper are as follows.

- We investigate the distinct strengths and optimization directions of various optimizers across different tasks and parameter states. Furthermore, we demonstrate that combining different optimizers during training can help achieve higher-quality models and better convergence.

- We present a novel fine-grain optimizer switch method called Interleaving Optimizer for Model Training (IOMT), which automatically switches suitable optimizers according to the parameter state during training. IOMT estimates the performance of optimizers under different parameter states by constructing Gaussian surrogate models and calculates the optimization gain using the acquisition function. By iteratively selecting the optimizer with the highest gain score, IOMT produces higher-quality models with faster convergence.

- We implement IOMT and conduct experiments on multiple models and tasks, including full training and partial fine-tuning. The experimental results demonstrate the advantages of our methods, such as achieving over 1% improvement in predictive accuracy with 10% reduction in convergence time, while also yielding superior generalization models. In addition, the case study and several independent experiments are presented to further explore the performance of IOMT.

## 2 RELATED WORKS AND BACKGROUND

In this section, we provide the background of our work, including the optimizers and the hybrid optimizer methods. After that, we identify the limitations of existing approaches.

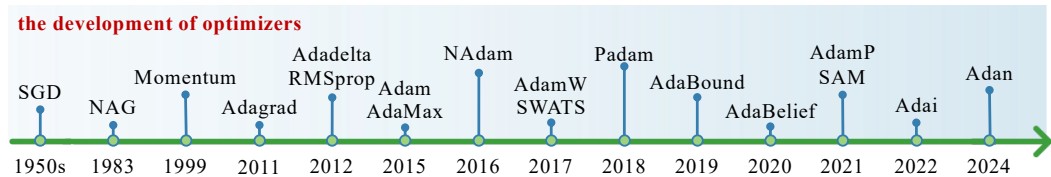

Figure 2: The development of neural network optimizers.

**Optimizers.** The optimizers and their hyperparameters are crucial for training DNNs, as they effectively adjust the model's parameters to minimize the loss function. The traditional gradient descent algorithm calculates the gradient of the loss function with respect to the model's parameters across the entire dataset and updates the parameters in the direction that reduces the loss (Ruder, 2016). Following the gradient descent algorithm, researchers have proposed a variety of optimizers. Figure 2 illustrates a portion of the historical development of these optimizers. Instead of calculating the gradient using the entire dataset, the Stochastic Gradient Descent (SGD) approximates the gradient by using only a single sample or a small batch of samples (Robbins & Monro, 1951). To address the slow convergence in ravines, the momentum technique is introduced in SGD (Sutskever et al., 2013). The Nesterov Accelerated Gradient (NAG) further enhances convergence speed and accuracy by incorporating a look-ahead mechanism into the update process (Qu & Li, 2019). Additionally, researchers have explored methods for adaptive learning rates based on different model parameters, such as RMSProp (Graves, 2013), Adam (Kingma & Ba, 2014), and AdamW (Loshchilov & Hutter, 2017). Beyond these, researchers have also proposed various second-order optimizers, such as L-BFGS (Liu & Nocedal, 1989), K-FAC (Martens & Grosse, 2015), and AdaHessian (Yao et al., 2021)). However, due to their practical application challenges, second-order optimizers are not further discussed in this paper. Additionally, researchers have attempted to develop new neural network-based learned optimizers through a meta-learning approach (Andrychowicz et al., 2016; Harrison et al., 2022).

**Hybrid optimizer.** Like other hyperparameter settings in training, there is no universal optimal optimizer in practical training (Wilson et al., 2017). For instance, SGD with momentum is commonly used in Computer Vision (CV), while Adam is favored for training transformer models in Natural Language Processing (NLP) (Yao et al., 2021). Some researchers have explored the performance of different optimizers during training, noting that different optimizers follow distinct descent paths at different saddle points (Im et al., 2016). Leveraging insights from multiple optimizers during model training is crucial in both academic research and practical applications. While numerous studies have investigated the adjustment of learning rates within optimizers (Gotmare et al., 2018; He et al., 2016; Smith, 2017), research on switching between different optimizers remains limited. Existing studies primarily focus on the basic form of switching, which involves transitioning from one optimizer to another. For example, SWATS (Keskar & Socher, 2017) achieves favorable results by initially using Adam and then switching to SGD. Padam (Chen et al., 2018) introduces a partial adaptive parameter to integrate Adam with SGD. Meanwhile, AdaBound (Luo et al., 2019) implements dynamic bounds on learning rates to facilitate a gradual and smooth transition.

**Limitations of current approaches.** (i) Single optimizer: although researchers are continually enhancing existing optimizers to better adapt to model parameter states (e.g., ravines), the associated computational cost cannot be ignored. In practical training, these complex optimizers do not necessarily outperform basic SGD (Keskar & Socher, 2017). To obtain better models, researchers need to train with different optimizers, which is a time-consuming process. Additionally, consistent optimizer training throughout the entire process (i.e., coarse-grain training) limits both model quality and convergence speed. (ii) Hybrid optimizer: combining the advantages of different optimizers can help improve model quality and convergence speed. Existing methods are limited to adjusting learning rates or transitioning between two types of optimizers, neglecting the unique strengths of different optimizers under different parameter states. Such a coarse mixing approach not only restricts the stability of the model quality but also impacts convergence speed (Zhuang et al., 2020).

## 3 OUR PROPOSED METHODS: IOMT

To better utilize multiple optimizers, we propose a novel fine-grain optimizer switch method called **I**nterleaving **O**ptimiezer for **M**odel **T**raing (IOMT), which enables adaptive optimizer switching during model training. In this section, we first provide a brief overview of IOMT. Then, we offer a detailed introduction including its problem formulation, surrogate model, and acquisition function.

### 3.1 OVERVIEW OF OUR PROPOSED IOMT

Figure 3 illustrates the workflow of IOMT, and a detailed description of IOMT with its pseudocode is presented in Appendix A. IOMT calculates the transferability weight $\omega_t$ to assist in the subsequent selection of optimizers before the training (Step 1). During each optimizer switch cycle (i.e., a few iterations), IOMT first compresses the model parameters $\theta^i$ to get the input of the surrogate model (Step 2). Then, IOMT selects the appropriated optimizer $o^i$ for the training of the current stage (Steps 3-4). Obtaining the training losses, IOMT calculates the performance score $s$ and updates the corresponding surrogate model $g^i$ (Steps 5-6). By iteratively executing this process, IOMT achieves the fine-grain optimizer switching.

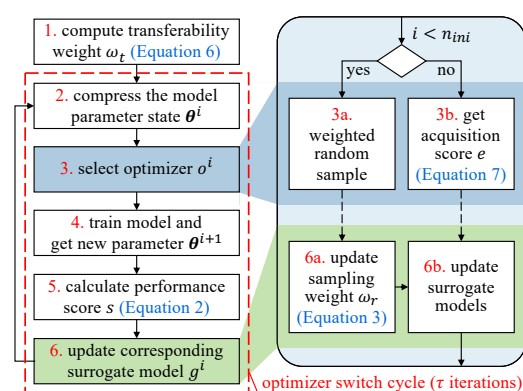

Figure 3: The workflow of IOMT.

For selecting the next optimizer (Step 3), IOMT employs two methods: the weighted random selection and recommendation based on the surrogate model. In the initial training stages, IOMT uses the calculated score $s$ to update the sampling weight $\omega_r$ for randomly selecting optimizers (Steps 3a and 6a). After acquiring enough training results, IOMT selects the optimizers with the highest gain $e$ calculated by the acquisition function for each training stage (Step 3b).

### 3.2 MOTIVATION AND PROBLEM FORMULATION

Before introducing the details of the surrogate model and acquisition function in IOMT, we first provide the hypothesis underlying our method: *"different optimizers offer distinct optimization directions and are suited to different parameter states"*. Figure 4 illustrates four examples of the different optimization directions, which correspond to the subfigures in Figure 1. It can be observed that although the five optimizers provide similar directions at the first iteration, their optimization paths diverge significantly after a few iterations. Previous studies have also observed this phenomenon, noting that optimizers exhibit different optimization directions under varying parameter states from both theoretical and visualization perspectives (Im et al., 2016). Therefore, we think that

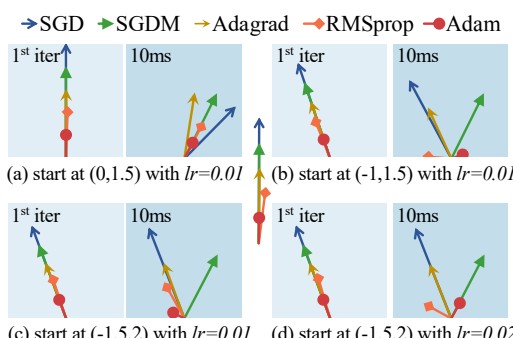

Figure 4: The illustration of different optimizer directions from the same start point.

the category of optimizers, like other hyperparameters, requires fine-grain tuning (i.e., dynamic algorithm configuration) (Adriaensen et al., 2022).

Building on this assumption, IOMT attempts to propose a fine-grain optimizer switch method that leverages the strengths of different optimizers for distinct parameter states. Let $o \in \mathcal{O}$, $\boldsymbol{\lambda} \in \Lambda$, and $t \in \mathcal{T}$ denote the optimizer type (e.g., SGD), hyperparameter setting (e.g., learning rate as 0.1) and the training time (e.g., 5 iterations), respectively. Then, the training process with fine-grain optimizer switches can be defined by a list of configurations $\mathcal{C} = \{c^1, c^2, ..., c^n\}$ where $c^i = (o^i, \boldsymbol{\lambda}^i, t^i)$. The objective of IOMT is to find an optimal $\mathcal{C}^*$ that minimizes the following objective function:

$$\mathcal{C}^* = \underset{\mathcal{C} \in \mathcal{O} \times \Lambda \times \mathcal{T}}{\arg\min} \mathcal{L}(\boldsymbol{\theta}^0, \mathcal{M}, \mathcal{D}, \mathcal{C}) \tag{1}$$

where $\boldsymbol{\theta}^0$ is the initial model parameter state, $\mathcal{L}(\cdot)$ denotes the loss of the trained model $\mathcal{M}$ under dataset $\mathcal{D}$. Equation (1) can be interpreted as fine-grain optimizer tuning for neural network training. When all $c^i \in \mathcal{C}$ share the same settings, it aligns with the traditional training process, which is described further in Section 4.1. For clarity, in the following sections, we set all training times $t \in \mathcal{T}$ to a specific value $\tau$, such as 5 iterations.

### 3.3 ESTIMATING OPTIMIZATION PERFORMANCE WITH SURROGATE MODELS

IOMT employs a Sequential Model-Based Optimization (SMBO) to address this fine-grain optimizer tuning problem, as illustrated in Figure 5. Initially, IOMT trains the model $\mathcal{M}$ using random configurations to obtain training experience (the blue block). Next, IOMT constructs surrogate models $\mathcal{G} = \{g_1, g_2, ..., g_m\}$ for each optimizer type $o_i \in \mathcal{O} = \{o_1, o_2, ..., o_m\}$ to guide the selection of suitable configurations (the red block). By iteratively selecting training configurations and updating surrogate models, IOMT achieves a fine-grain optimizer switch training. In this section, we introduce IOMT's surrogate model through its selection, input, output, and initialization.

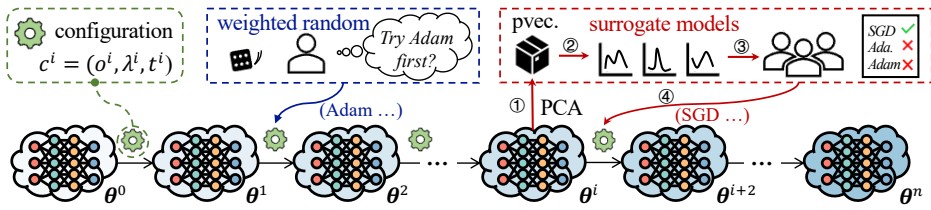

Figure 5: The training process of our proposed IOMT.

**The selection of the surrogate model.** IOMT utilizes the Gaussian process (GP) model (Schulz et al., 2018) as its surrogate model for several reasons. First, compared to other machine learning

models, GPs can efficiently train and continuously update the surrogated model as the training progresses. Second, GPs provide uncertainty estimation for predictions (i.e., the variance information), which is useful for guiding the optimizer selection (as detailed in Section 3.4). Thirdly, as a powerful probabilistic model, GPs effectively construct the overall distribution based on known points, offering good flexibility and interpretability.

**The input of the surrogate model.** At the beginning of each optimizer cycle, IOMT acquires the input for the surrogate model $\text{VEC}^i$. The traditional surrogate model in SMBO uses the hyperparameter $\boldsymbol{\lambda}^i$ as its inputs. In IOMT, the input $\text{VEC}^i$ also includes a vector representing the parameter state $\boldsymbol{\theta}^i$ to learn the impact under different parameter states. Considering the high cost of using the full model parameters, IOMT applies feature engineering to reduce the input size. Specifically, IOMT uses Principal Component Analysis (PCA) (Labrín & Urdinez, 2020) to compress the parameters layer by layer, lowering the training cost for the surrogate model. To further reduce the training cost of surrogate models during training, IOMT selects only a few layers of the model as inputs for the surrogate model (e.g., the classifier layer with a few hidden layers). In the case of partial fine-tuning, IOMT focuses solely on the trainable parameters (e.g., the matrices A and B in LoRA).

**The output of the surrogate model.** In contrast to the results obtained from training to convergence, IOMT emphasizes the "short-term" benefits each optimizer can achieve given the current parameter state. Therefore, the output of IOMT's surrogate model does not use the final loss or accuracy, but instead employs a computed performance score $s \in [-1, 1]$. During the training of a stage, IOMT performs multiple iterations of training, resulting in a set of losses, denoted as $\boldsymbol{l} = \{l_1, l_2, ..., l_\tau\}$, and $l_0$ represents the loss before training. IOMT first calculates the loss variation $\Delta l_i = \frac{l_{i-1}-l_i}{\max(l_i, l_{i-1})}$ for each iteration to get the average reduction $\mu_\Delta$ and variance $\sigma_\Delta$. To estimate the optimization performance of different configurations, IOMT combines the considerations of exploration (i.e., variance $\sigma_\Delta$) and exploitation (i.e., mean $\mu_\Delta$) to calculate a weighted score $s = \mu_\Delta + \alpha\sigma_\Delta$. However, such a weighted score overlooks the direction of variance. For instance, in Figure 6(a), optimizers $o_1$ and $o_3$ have the same mean $\mu_\Delta$ and variance $\sigma_\Delta$, yet $o_3$ achieves a lower loss than $o_1$ during training. A similar issue arises in the comparison between $o_2$ with $o_1$ and $o_3$. To address this problem, we incorporate boundary considerations into the performance calculation, including the upper bound $\Delta_{\text{UPPER}} = \frac{l_0-\max(\boldsymbol{l})}{\max(l_0,\max(\boldsymbol{l}))\times\tau}$ and lower bound $\Delta_{\text{LOWER}} = \frac{l_0-\min(\boldsymbol{l})}{\max(l_0,\min(\boldsymbol{l}))\times\tau}$. Then, the final optimization performance score is defined as follows,

$$s = \tanh(\frac{1}{3}(\mu_\Delta + \Delta_{\text{UPPER}} + \Delta_{\text{LOWER}}) + \alpha\sigma_\Delta) \tag{2}$$

where $\alpha$ represents the weight for variance.

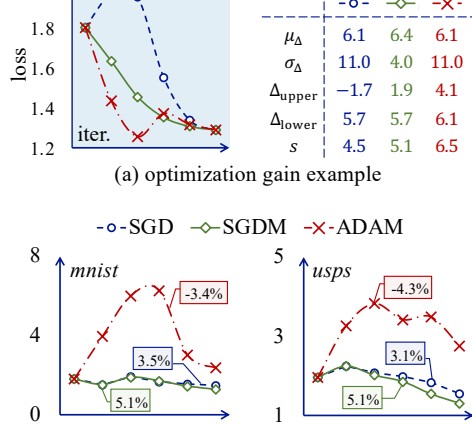

(a) optimization gain example

(b) two practical examples

Figure 6: Examples of the optimization gain.

**The initial weighted random selection.** To obtain enough training experience for the construction of surrogate models, IOMT trains with random configurations at the start of training. Though randomly selecting configurations for initial training can yield the necessary experience, IOMT employs a weighted random initialization method to enhance the performance of the initial training. Specifically, IOMT maintains a sampling weight $\boldsymbol{\omega}_r[j] \in [0, 1]$ for each type of optimizer $o_j$ and its surrogate model $g_j$, presenting the probability of being sampled. This sampling weight is initially assigned a value of 1 to achieve a random initialization. After completing the training with the current configuration, the sampling weight for the corresponding optimizer $\omega_j$ is updated to the normalized optimization performance score as shown in Equation (3), where $\omega_{\min}$ represents the minimal threshold.

$$\boldsymbol{\omega}_r[j] = \max(\frac{1}{2}(s+1), \omega_{\min}). \tag{3}$$

## 3.4 Selecting Optimizers with Acquisition Function

Although the calculated optimization performance $s$ can be used to select configurations directly, given the volatility of the loss and the complexity of model training, IOMT considers additional factors in the design of acquisition, including variance, transferability, and the training process. In this section, we introduce considerations designed for the acquisition function used in IOMT.

**Consideration of variance.** Benefiting from the advantages of the Gaussian process model, the surrogate model can provide both the mean score $s_\mu$ and an estimate of the variance $s_\sigma$. Similar to traditional hyperparameter optimization methods, IOMT also incorporates a trade-off between exploration and exploitation in the acquisition function as follows

$$\text{ACQ}(s_\mu, s_\sigma) = s_\mu + \alpha s_\sigma, \tag{4}$$

where $\alpha$ represents the weight for variance, consistent with the definition in Equation (2).

**Consideration of transferability.** The training cost of DNNs is closely related to the initial model parameter state $\theta^0$. In fine-tuning, closely related upstream and downstream tasks (i.e., high transferability between the pre-trained model and the new task) are easier to train than those that are dissimilar. Considering the idea that *"a pre-trained model with lower transferability necessitates more substantial tuning adjustments"*, we use the model's transferability $\omega_t$ as the weight of the variance in the acquisition function, as shown in the following equation,

$$\text{ACQ}(s_\mu, s_\sigma, \omega_t) = s_\mu + (1 - \omega_t)s_\sigma. \tag{5}$$

The transferability $\omega_t$ is calculated using two types of evaluation metrics, including performance-based metric $\omega_p$ and distribution-based metrics $\omega_d$. Firstly, the performance-based metric $\omega_p \in [0, 1]$ is the testing result (e.g., accuracy) which is directly tested with the pre-trained model without deep refining. Meanwhile, IOMT also uses some feature-based metrics, which analyze the distribution of the output vectors or labels, to estimate the model's transferability, including LogME (You et al., 2021) and Leep (Nguyen et al., 2020). Equation (6) presents the definition of transferability weight.

$$\omega_t = \beta\omega_p + (1 - \beta)\frac{1}{k}\sum_{i=1}^{k} \text{sigmoid}(\omega_d^i). \tag{6}$$

where $\omega_d^i$ represents $k$ distribution-based metrics and $\beta$ represents the weight for two kinds of metrics. We use the sigmoid function to constrain the distribution-based metric within the range of $[0, 1]$ to align with the performance-based metric. Then, the weighted sum reflects the transferability of the initial model for current tasks. A higher transferability weight indicates higher transferability, while a lower one suggests lower transferability.

**Consideration of training process.** Additionally, IOMT takes into account the differing needs in the early and later stages of training, specifically that *"after the model becomes stable, smaller tuning adjustments are needed."* As training progresses, the model continuously captures the knowledge required for the current task, leading to a stabilization of the training loss. At the later stages of the training, the target position on the parameter surface is constrained within a smaller range. In this context, optimizers with larger amplitudes may disrupt the tuning process. Therefore, the proportion of variance in the acquisition function should be reduced. Hence, we introduce a periodic halving of the weight for variance information in IOMT as Equation (7), where $i$ represents the current iteration and $n$ represents the halving period.

$$e = \text{sigmoid}(s_\mu + (1 - 2^{-\lfloor i/n \rfloor} \cdot \omega_t)s_\sigma). \tag{7}$$

## 4 Discussion

To further introduce our proposed IOMT, we discuss its differences from the hyperparameter tuning (HPO) and SMBO, along with its advantages and limitations in this section.

### 4.1 Analyzing the Differences between IOMT and HPO

**Compare with HPO.** The optimizer, as one of the hyperparameters in DNNs, its automatic adjustment is a form of HPO and AutoML. However, the vanilla training addresses it as a coarse-grain

HPO, where the hyperparameters remain fixed throughout the whole training process. The optimization objective of such coarse-grain tuning can be formulated as below,

$$c^* = \underset{c \in \mathcal{O} \times \Lambda \times \mathcal{T}}{\arg\min} \mathcal{L}(\boldsymbol{\theta}^0, \mathcal{D}, c) \tag{8}$$

where $c = (o, \lambda, t)$ represents the hyperparameter configurations (same as the defination in Section 3.2). Compared to IOMT's fine-grain tuning (i.e., Equation 1), the vanilla HPO restricts the way model parameters are updated and the collaboration among different optimizers. Additionally, though researchers have proposed hybrid methods that combine binary optimizers, these approaches still integrate the optimizers from rules of thumb rather than performing fine-grain hyperparameter optimization. For example, SWATS (Keskar & Socher, 2017) switches the training from Adam to SGD based on the foundation that "Adam quickly adapts to problems in the early training phase, while SGD promotes better generalization in the later stages".

**Compare with SMBO.** IOMT adopts the idea of surrogate models and the acquisition function in SMBO, but there are significant differences between IOMT and SMBO. First, the SMBO only considers the impact of hyperparameters on the results, neglecting changes in the model parameter states. When the initial parameter states differ, the performance evaluation of hyperparameters is also different. In contrast, IOMT introduces additional parameter inputs to the surrogate model and considers the training progress in the acquisition function to study the "short-term" gain on different parameter states. Secondly, SMBO aims to select the best hyperparameters (i.e., coarse-grain tuning), whereas IOMT aims to obtain the best model (i.e., fine-grain tuning). Compared to SMBO, IOMT enables the collaboration of various hyperparameters within a single training process.

## 4.2 ANALYZING THE ADVANTAGES OF IOMT

**Accuracy:** IOMT achieves a DNN training with interleaving optimizers, enabling collaboration among multiple optimizers. This fine-grain optimizer tuning not only integrates the optimization strategies of different optimizers but may also yield an optimization path (i.e., the final trained model) that a single optimizer cannot achieve, resulting in higher accuracy. Figure 7(a) provides examples across three different functions, illustrating that IOMT can discover optimization paths that a single optimizer cannot achieve. Similarly, the final model weights obtained from training for the same number of epochs on the *cifar10* dataset using different optimizers show significant differences, as illustrated in Figure 7(b). This hybrid approach, which employs multiple optimizers, expands the search space of traditional training, leading to an improved accuracy upper bound.

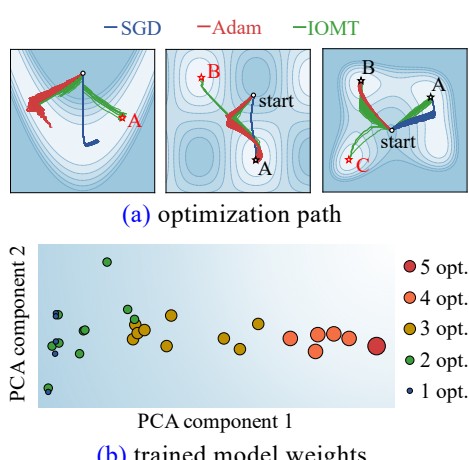

(a) optimization path

(b) trained model weights

Figure 7: Analyze of interleaving training.

**Training efficiency:** We analyze the time cost of IOMT using the training of ResNet18 (whose training FLOPs $t_M \approx 1.8 \times 10^9$) on the *cifar10* dataset (i.e., feature dimensions $D \approx 3 \times 10^3$, instance number $N = 6 \times 10^4$, and class number $K = 10$) with epoch number $n_{\text{EPOCH}} = 100$, batch size $n_{\text{BZ}} = 64$ and switching iteration number $\tau = 20$ as an example. For the vanilla training, the time cost for a single epoch is $t_{\text{TRAIN}} \approx 2 \times t_M \times N \approx 2 \times 10^{14}$. Compared to vanilla training, IOMT incurs additional time consumption due to two processes: transferability assessment before training $t_{\text{EST}}$ and the updating of the surrogate model during the training process. First, the $t_{\text{EST}}$ includes the computation for two distribution-based metrics (i.e., LEEP and LogME) and one performance-based metric. Among them, the time cost for the LEEP and performance-based metrics is equivalent to a single forward pass (Nguyen et al., 2020), while the computational complexity of LogME is $O(KD^2 + NKD + D^3 + ND^2) \approx 3 \times 10^{10}$ (You et al., 2021). Then, the transferability assessment time $t_{\text{EST}} \approx t_{\text{TRAIN}}$ (actually smaller in practical, e.g., $t_{\text{EST}} = t_{\text{TRAIN}} \times 1\%$). Second, the additional time consumption from the updating of the surrogate model $t_{\text{SUR}}$ includes the PCA compression of the selected parameters and the updating of the Gaussian process model. The time complexity of compression and updating is $O(W^2D')$ and $O(N_{sw}^3)$, where $W \approx 10^4$ represents the number of selected parameters (i.e., only the last layer), $D' \approx 100$ represents the number of PCA components,

Table 1: Test accuracy (%) of the full training with different optimizers.

| method | usps | | mnist | | stl10 | | cifar10 | |
|---|---|---|---|---|---|---|---|---|
| | ResNet18 | ViT | ResNet18 | ViT | ResNet18 | ViT | ResNet18 | ViT |
| SGD | $96.10_{\pm0.24}$ | $97.46_{\pm0.70}$ | $99.29_{\pm0.03}$ | $99.50_{\pm0.04}$ | $86.94_{\pm0.49}$ | $97.83_{\pm0.22}$ | $80.73_{\pm0.43}$ | $97.53_{\pm0.03}$ |
| SGDM | $95.83_{\pm0.30}$ | $97.68_{\pm0.11}$ | $99.47_{\pm0.05}$ | $99.65_{\pm0.04}$ | $86.99_{\pm0.36}$ | $96.61_{\pm0.04}$ | $81.64_{\pm0.60}$ | $97.58_{\pm0.04}$ |
| Adagrad | $96.00_{\pm0.94}$ | $93.40_{\pm0.04}$ | $99.40_{\pm0.06}$ | $98.24_{\pm0.50}$ | $83.38_{\pm7.99}$ | $78.66_{\pm2.54}$ | $80.57_{\pm0.14}$ | $60.95_{\pm0.62}$ |
| RMSprop | $95.30_{\pm0.55}$ | $95.25_{\pm0.04}$ | $99.13_{\pm0.17}$ | $98.14_{\pm0.09}$ | $69.91_{\pm2.06}$ | $88.62_{\pm4.45}$ | $71.92_{\pm0.41}$ | $78.09_{\pm0.92}$ |
| Adam | $95.13_{\pm0.53}$ | $93.26_{\pm0.78}$ | $99.11_{\pm0.06}$ | $99.01_{\pm0.08}$ | $76.49_{\pm2.04}$ | $82.26_{\pm1.16}$ | $72.33_{\pm0.84}$ | $75.23_{\pm0.92}$ |
| SWATS | $95.53_{\pm0.71}$ | $94.00_{\pm1.23}$ | $99.17_{\pm0.11}$ | $98.73_{\pm0.13}$ | $79.76_{\pm2.11}$ | $88.03_{\pm0.44}$ | $75.17_{\pm0.21}$ | $66.15_{\pm4.74}$ |
| Padam | $96.10_{\pm0.15}$ | $97.58_{\pm0.11}$ | $99.46_{\pm0.02}$ | $99.66_{\pm0.04}$ | $85.64_{\pm0.48}$ | $90.81_{\pm0.06}$ | $81.58_{\pm0.38}$ | $96.03_{\pm0.03}$ |
| AdaBound | $95.02_{\pm0.17}$ | $87.64_{\pm0.84}$ | $99.25_{\pm0.06}$ | $97.54_{\pm0.13}$ | $84.48_{\pm0.42}$ | $86.33_{\pm2.39}$ | $69.27_{\pm5.02}$ | $70.91_{\pm4.35}$ |
| ours | $\mathbf{96.81_{\pm0.21}}$ | $\mathbf{97.81_{\pm0.21}}$ | $\mathbf{99.51_{\pm0.01}}$ | $\mathbf{99.71_{\pm0.02}}$ | $\mathbf{88.23_{\pm0.23}}$ | $\mathbf{98.21_{\pm0.19}}$ | $\mathbf{84.14_{\pm0.11}}$ | $\mathbf{98.04_{\pm0.03}}$ |

and $N_{sw} = n_{\text{EPOCH}} \frac{N}{\tau \times n_{\text{BZ}}} \approx 5 \times 10^3$ represents the total switching operations in the tuning process. Then, we can calculate $t_{\text{SUR}} \approx 10^9 \ll t_{\text{TRAIN}}$. Since $t_{\text{EST}}$ is executed only once before training and $t_{\text{SUR}} \ll t_{\text{TRAIN}}$, the additional time in IOMT is minimal. Furthermore, thanks to its adaptability to different parameter states, IOMT is able to achieve better convergence speed.

## 5 EXPERIMENTAL STUDY

To investigate the rationality of IOMT, we conducted experiments and present the experimental results in this section. We first exhibit two overall experiments to observe the performance of IOMT in full training and PEFT. Then, we illustrate a case study to observe the practical switching process of IOMT during training. In addition, several independent experiments are presented to investigate the significance of designs within IOMT.

In the experiments, we used 4 ImageNet pre-trained models available from PyTorch (Paszke et al., 2019) (i.e., ResNet18, ResNet152, MobileNet V2, and ViT) and 2 pre-trained NLP models from HuggingFace (Wolf et al., 2020) (i.e., RoBerta and LLaMA-7B). For the selection of datasets, we took 4 commonly used CV datasets from PyTorch (i.e., *usps*, *mnist*, *stl10*, and *cifar10*) and 3 NLP tasks from Hugging Face (i.e., *mrpc*, *qqp*, and *wnli*). In addition, the experiments were conducted on a Linux machine with a 128-core 2.6GHz Intel(R) Xeon(R) Platinum 8358 CPU and 512GB main memory. More details of the models and datasets used in our experiments can be found in Appendix B.

### 5.1 OVERALL PERFORMANCE OF IOMT

We first compared our proposed IOMT with the training using a single optimizer or hybrid optimizers under full training and PEFT. Specifically, five commonly used optimizers were tested for single optimizer training: SGD (Robbins & Monro, 1951), SGDM (Sutskever et al., 2013), Adagrad (Duchi et al., 2011), RMSProp (Graves, 2013), and Adam (Kingma & Ba, 2014). For hybrid optimizer training, we included SWATS (Keskar & Socher, 2017), Padam (Chen et al., 2018), and AdaBound (Luo et al., 2019). The initial learning rate and training epochs of each method were setting as [0.1,0.01,0.001] and 100. For IOMT, we set the initial steps $n_{ini} = 50$ and training time $\tau = 25$ iterations. More details of the baselines and settings are presented in Appendix C.

**Experiments on full training.** The experimental results show that the switching method proposed in IOMT can always achieve good improvements in

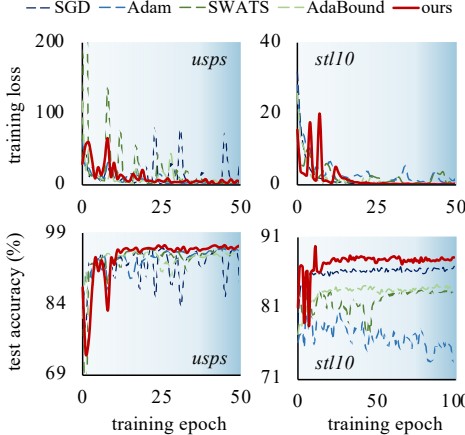

Figure 8: The training loss and test accuracy line graph.

test accuracy (i.e., 1%-3%), as shown in Table 1. However, other hybrid methods often perform worse than training with a single optimizer, especially in complex tasks (i.e., *stl10* and *cifar10* datasets). Additionally, compared to other methods, IOMT exhibits smaller variance, indicating more stable performance outputs. To illustrate the convergence of IOMT, we present the training loss and test accuracy of each method in the Figure 8. For ease of observation, the baselines with significant fluctuations are not displayed in the figure. It can be observed that IOMT shows a faster convergence speed compared to the vanilla method.

**Experiments on PEFT.** In addition to the full training, we also compared the proposed IOMT with baselines on the PEFT that only update partial of the model parameters, including the head fine-tuning (Poojary & Pai, 2019) in CV problems and the LoRA (Hu et al., 2021) in NLP tasks. To analyze the convergence performance, we terminated the training when the convergence conditions were satisfied, i.e., the change of loss is less than $1 \times 10^{-4}$ in 10 consecutive epochs or the training reaches 100 epochs. Table 2 presents a partial of the experimental results, more experimental results and setting details can be found in the Appendix C. Like its performance in full training, IOMT achieves higher accuracy and $F_1$ score (up to 2%) for both CV and NLP tasks. In terms of convergence time, the end-to-end results shown in the table indicate that IOMT has a faster convergence speed in PEFT (e.g., 10% faster on *usps*. Meanwhile, the time cost for transferability assessment (i.e., the time indicated after "+" in the table) is much smaller than the training time, which is consistent with the discussion in Section 4.2.

Table 2: Test accuracy (%), $F_1$ score (%) and convergence time (sec.) of the PEFT. ViT for the CV datasets (i.e., *usps* and *stl10*) and RoBerta for the NLP datasets (i.e., *mrpc* and *qqp*).

| method | usps | | stl10 | | mrpc | | qqp | |
|---|---|---|---|---|---|---|---|---|
| | accuracy | time | accuracy | time | accuracy | $F_1$ score | accuracy | $F_1$ score |
| SGD | $94.42_{\pm0.21}$ | 3169 | $97.75_{\pm0.16}$ | 275 | $85.21_{\pm0.35}$ | $87.21_{\pm0.32}$ | $82.13_{\pm0.52}$ | $75.19_{\pm0.82}$ |
| SGDM | $95.67_{\pm0.14}$ | 2397 | $98.37_{\pm0.10}$ | 483 | $85.54_{\pm0.69}$ | $86.27_{\pm0.41}$ | $83.30_{\pm0.63}$ | $75.27_{\pm0.82}$ |
| Adagrad | $95.37_{\pm0.21}$ | 2220 | $98.34_{\pm0.09}$ | 284 | $84.94_{\pm0.59}$ | $87.29_{\pm0.46}$ | $83.47_{\pm0.79}$ | $73.28_{\pm0.83}$ |
| RMSprop | $94.64_{\pm0.53}$ | 2208 | $97.91_{\pm0.07}$ | 568 | $84.09_{\pm0.76}$ | $89.24_{\pm0.74}$ | $82.09_{\pm0.63}$ | $74.09_{\pm0.92}$ |
| Adam | $94.47_{\pm0.49}$ | 2215 | $98.36_{\pm0.03}$ | 694 | $86.52_{\pm0.71}$ | $90.37_{\pm0.92}$ | $82.27_{\pm0.71}$ | $74.92_{\pm0.84}$ |
| SWATS | $95.12_{\pm0.21}$ | 2643 | $98.38_{\pm0.10}$ | 822 | $86.27_{\pm0.62}$ | $90.34_{\pm0.42}$ | $80.79_{\pm0.81}$ | $74.80_{\pm0.19}$ |
| Padam | $95.72_{\pm0.42}$ | 2234 | $98.38_{\pm0.10}$ | 598 | $80.64_{\pm0.32}$ | $87.07_{\pm0.82}$ | $73.04_{\pm0.91}$ | $80.93_{\pm0.83}$ |
| AdaBound | $95.42_{\pm0.11}$ | 2232 | $98.30_{\pm0.14}$ | 199 | $68.38_{\pm0.59}$ | $81.22_{\pm0.94}$ | $78.64_{\pm0.49}$ | $79.01_{\pm0.42}$ |
| ours | $\mathbf{96.12_{\pm0.10}}$ | $\mathbf{2030+2}$ | $\mathbf{99.01_{\pm0.09}}$ | $\mathbf{180+1}$ | $\mathbf{87.99_{\pm0.13}}$ | $\mathbf{91.36_{\pm0.15}}$ | $\mathbf{85.57_{\pm0.14}}$ | $\mathbf{81.18_{\pm0.31}}$ |

In summary, IOMT demonstrates excellent tuning performance and convergence speed across different training approaches, various models, and downstream tasks. The combination of model transferability analysis and optimizer switching based on parameter surface characteristics effectively assists DNN training.

## 5.2 CASE STUDY FOR THE SWITCHING PROCESS OF IOMT

To observe IOMT's switching process, we conducted a case study with a simple task *hymenoptera* from Kaggle and a restricted optimizer space (only for SGD and SGDM). The training loss and the optimizer switch process are plotted in Figure 9. After the initial stages with weighted random sampling, IOMT selects the suitable optimizer with faster convergence speed for training, i.e., the SGDM selected in Figure 9. After that, the optimizer switch operation occurs when a decrease in the convergence speed of the optimizer (Point A) or detects a local stable state (Point C). Additionally,

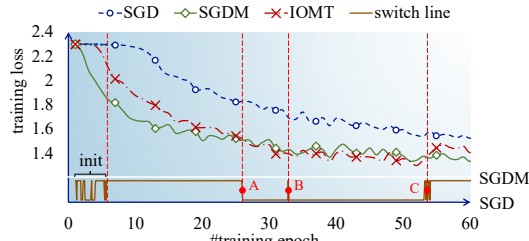

Figure 9: The training loss line of the case study with vanilla FT and IOMT.

during tuning, IOMT may also undergo temporary switches to adjust the optimization state (Point B). This case study demonstrates that IOMT can effectively select the appropriate optimizer based on the model parameter state, thereby improving convergence speed and model quality.

## 5.3 INDEPENDENT EXPERIMENTS

Additionally, we conducted several independent experiments to further analyze the effectiveness of IOMT. In this section, we outline the main conclusions, with further details available in Appendix D.

**The optimizer selection strategy.** IOMT employs an optimizer selection strategy that considers variance, transferability, and training process. Table 7 presents comparative results for different selection strategies. Compared to random or periodic switching, IOMT achieves higher accuracy (up to 2%) and lower variance. Additionally, the ablation experimental results indicate that the designs for transferability assessment and variance reduction further enhance its advantages.

**The initial selection method.** Compared with random selection, the weighted selection in IOMT significantly enhances the stability of the surrogate model, which reduces variability in the training outcomes, as shown in Figure 10(a).

**The model compression technique.** Table 10 illustrates the effects of various feature compression techniques on training results. For the selected tasks (i.e., *usps* and *mnist*), simple methods like random projection and PCA outperform the more complex UMAP technique. This suggests that basic compression techniques are adequate for training the surrogate model.

**The optimizer search space.** We broadened the hyperparameter space of candidate optimizers to explore how this expanded search space affects IOMT's performance. The experimental results shown in Figure 11 indicate that IOMT continues to perform well in the enlarged search space.

**The influence of hyperparameter setting.** We also performed an experimental analysis on the hyperparameters in IOMT, including the initial step $n_{ini}$, switch step size $\tau$, and the number of PCA components. Figure 10(b-d) presents the experimental results, demonstrating that a small initial step (e.g., only 10 for small dataset *usps*), switch step size (10% of an epoch) and PCA components (e.g., 2) can achieve good accuracy. A more detailed analysis can be found in Appendix D.

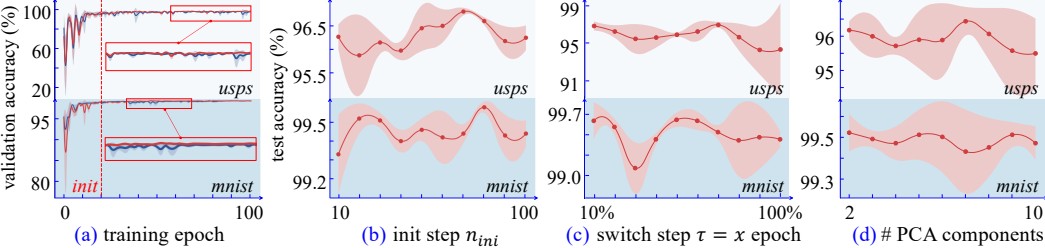

Figure 10: The experimental results of independent experiments.

## 6 CONCLUSION

The selection of optimizers and their hyperparameters plays a crucial role in deep neural network (DNN) training. Traditionally, researchers use a single optimizer during the whole training (i.e., a coarse-grain optimizer tuning), which limits the model quality and convergence speed. Currently, some works attempt to leverage the advantages of different optimizers during training to achieve higher-quality models. However, these methods are still constrained by merely adjusting the learning rate or transitioning between two types of optimizers, overlooking the unique strengths of various optimizers under different parameter states. To better combine the benefits of different optimizers, we introduce a fine-grain optimizer switch method called Interleaving Optimizers for Model Training (IOMT) in this paper. Specifically, IOMT constructs surrogate models during training to estimate the performance of different optimizers under varying model parameter states. In addition, IOMT employs a transferability assessment to enhance the selection of optimizers. Combining the predicted performance and transferability information with an acquisition function, IOMT gets the estimation of optimization gain for each optimizer and switches the optimizer with the largest score for the training stage. The experimental results on full training and PEFT demonstrate that IOMT achieves a better model quality (e.g., 3% accuracy improvement on *stl10* dataset) with faster convergence (e.g., 10% on the *stl10* dataset). In addition, a case study and two independent experiments further investigate the optimizer switching process and design details of IOMT.

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

## A  OVERALL ALGORITHM AND PSEUDOCODE OF IOMT

For ease of reading, Table 3 provides the explanations of key notations used in this paper for IOMT.

In this section, we provide an overview of IOMT with the pseudocode in Algorithm 1. For ease of understanding, we only switch the types of optimizers in the description while keeping the hyperparameters and training time constant (i.e., $t^i = \tau$ iterations). Before the model's training, IOMT first calculates the transferability weight $\omega_t$ to assist in the subsequent selection of optimizers (Line 3). Then, a randomly selected initial optimizer $o_j$ is used for model training and loss calculation (Lines 8-9). When the number of iterations meets the time constant $\tau$, IOMT calculates the

Table 3: The description of notations used in IOMT.

| notation | description |
|---|---|
| switch step $\tau$ | the number of iterations for a single switch cycle. |
| init step $n_{ini}$ | the number of switch cycles for the initial phase. |
| selection weights $\omega_{\mathbf{r}}$ | the sample weights for the initial phase |
| transferability weight $\omega_t$ | the transferability weight which is calculated by the performance-based metric $\omega_t$ and distribution-based metric $\omega_d$ |

---

**Algorithm 1:** Basic framework of our proposed IOMT.

---

**Input:** Training Dataset $D = \{D_{train}, D_{test}\}$, $m$ optimizers $\mathcal{O} = \{o_1, ..., o_m\}$, the model for training
    $M$ with initial parameter state $\boldsymbol{\theta}^0$, the init steps $n_{ini}$, the switch step size $\tau$, the training epochs
    $n_{epoch}$

1 **initialize:** $n_t \leftarrow 0, n \leftarrow 0, \mathcal{G} \leftarrow \{g_1, g_2, ..., g_m\}, \boldsymbol{\omega}_r \leftarrow \{1|1, ..., m\}, j \leftarrow \text{RandomSelect}(m)$,
    $losses \leftarrow [], \boldsymbol{v} \leftarrow \text{CompressModel}(M, \boldsymbol{\theta}^0)$

2 /* calculate the transferability weight before the training         */

3 $\omega_t \leftarrow \text{CalculateTransferabilityWeight}(M, \boldsymbol{\theta}^0, D_{train})$   // as Equation (6)

4 /* training models with interleaving optimizers           */

5 **for** $i$ *in* $[1, n_{epoch}]$ **do**

6    **for** BATCH *in* $D_{train}$ **do**

7       $n \leftarrow n + 1$

8       $\boldsymbol{\theta}^n \leftarrow \text{TrainModel}(M, \boldsymbol{\theta}^{n-1}, \text{BATCH}, o_j)$

9       $l \leftarrow \text{CalculateLoss}(M, \boldsymbol{\theta}^n, \text{BATCH}), losses.\text{append}(l)$

10       **if** $n\%\tau = 0$ **then**

11          $s \leftarrow \text{CalculateOptimizationGain}(losses)$ // as Equation (2)

12          $g_j \leftarrow \text{UpdateSurrogateModel}(g_j, \boldsymbol{v}, s)$

13          $\boldsymbol{v} \leftarrow \text{CompressModel}(M, \boldsymbol{\theta}^n)$

14          // init steps with a weighted random selection

15          **if** $n_t < n_{ini}$ **then**

16             $\boldsymbol{\omega}_r[j] \leftarrow \text{UpdateSampleWeight}(s)$ // as Equation (3)

17             $j \leftarrow \text{WeightedRandomSelect}(\boldsymbol{\omega}_r)$

18          // following steps with a surrogated selection

19          **else**

20             $j \leftarrow \text{SurrogatedSelect}(\boldsymbol{v}, \omega_t, \mathcal{G}, \mathcal{O})$

21          **end**

22          $n_t \leftarrow n_t + 1, losses \leftarrow []$

23       **end**

24    **end**

25 **end**

**Output:** the trained model $M$ with parameter state $\boldsymbol{\theta}^n$

---

performance score $s$ based on all losses within time $t^i$ (Line 11). Subsequently, the current optimizer $o_j$ is updated by the surrogate model $g_j$ using the optimization gain $s$ and the model features $\boldsymbol{v}$ calculated at the end of the previous round (Lines 12-13). Based on the results from weighted random selection or surrogate model selection, IOMT obtains the suitable optimizer for the next training stage (Lines 14-20) and continues this process iteratively until the final training results $\boldsymbol{\theta}^n$ are achieved.

For the selection of the suitable optimizer, IOMT employs two types of methods: the weighted random selection and the surrogate model selection for the following steps. In the initial training steps (i.e., $n_t < \tau$), IOMT uses the optimization gain $s$ to update the sampling weight $\boldsymbol{\omega}_r$ for randomly select configurations for training (Lines 14-17). After obtaining enough training results (i.e., $n_t \geq \tau$), IOMT utilizes the trained surrogate models to select the configurations used for the following training (Line 20). The configuration with the highest score (i.e., Equation 7) is selected for the next training iteration.

## B    DATASETS AND MODELS USED IN EXPERIMENTS

In the experiments, we used 4 CV datasets from Pytorch (Paszke et al., 2019) (i.e., *usps*, *mnist*, *stl10*, and *cifar10*) and 2 NLP datasets from Hugging Face (Wolf et al., 2020) (i.e., *mrpc* and *qqp*). The information on these downstream tasks is as follows:

- *usps* (Hull, 1994): a classical digit dataset automatically scanned from envelopes by the U.S. Postal Service containing a total of 9,298 16×16 pixel grayscale samples, which includes 10 classes of figures.

- *mnist* (LeCun et al., 1998): a handwritten digits dataset with 28x28 grayscale figures, which has a training set of 60,000 examples and a test set of 10,000 examples.

- *stl10* (Coates et al., 2011): a 10-classes 96x96 color figure dataset, which has 500 training images and 800 test images per class. The dataset is inspired by the *cifar-10* (Krizhevsky et al., 2009) but with some modifications.

- *cifar10* (Krizhevsky et al., 2009): a 10-classes 32x32 color figure dataset, which has 5,000 training images and 1,000 test images per class.

- *mrpc* (Dolan & Brockett, 2005): the Microsoft Research Paraphrase Corpus, which consists of 5.8k sentence pairs that were automatically extracted from online news sources. The sentence pairs have been annotated by human raters to indicate whether the sentences within each pair are semantically equivalent.

- *qqp* (Quora): the Quora Question Pairs dataset, which consists of over 400,000 pairs of questions. Each question pair is annotated with a binary value indicating whether the two questions are paraphrases of each other.

As for the pre-trained models, we used 4 ImageNet pre-trained models available from Py-Torch (Paszke et al., 2019) (i.e., ResNet18, ResNet152, MobileNet V2, and ViT) and a pre-trained NLP models RoBerta (Camacho-collados et al., 2022) and LLaMA-7B (Touvron et al., 2023) from HuggingFace (Wolf et al., 2020) which trained on 124M tweets from January 2018 to December 2021, and finetuned for sentiment analysis with the TweetEval benchmark (Barbieri et al., 2020). It can be found that among the downstream tasks, *stl10*, *mrpc* and *qqp* are relatively close to the upstream task, and *usps* and *mnist* have a certain correlation with the upstream task. We selected downstream datasets with varying degrees of relevance to the upstream task, to better analyze the performance of the proposed method in different scenarios.

## C    DETAILS OF OVERALL EXPERIMENTS

### C.1    EXPERIMENT SETTINGS

In the overall experiment, we compared 5 single optimizer methods (i.e., SGD (Robbins & Monro, 1951), SGDM (Sutskever et al., 2013), Adagrad (Duchi et al., 2011), RMSProp (Graves, 2013) and Adam (Kingma & Ba, 2014)), 3 hybrid methods (i.e., SWATS (Keskar & Socher, 2017), Padam (Chen et al., 2018), and AdaBound (Luo et al., 2019)), and the proposed IOMT. The single methods are all from the PyTorch implementation, and except for the learning rate being set in [0.1,0.01,0.001] and the epoch number being set to 100, the other hyperparameters are set to the default values in PyTorch. Additionally, for the three hybrid methods, we installed and used the original implementations from the authors via Github and PyPI. The hyperparameter settings were kept at their defaults, except for the epoch number, which was adjusted to be consistent with the other methods. In addition, all the experiments in this paper are conducted 3 times with different random seeds to avoid randomness.

### C.2    MORE EXPERIMENTAL RESULTS

In addition to the results in Section 5.1, we also conducted more experiments to analyze the characteristics of IOMT, and the results are presented in this section.

We first compared IOMT with additional baselines, including various optimizers (i.e., ASGD (Polyak & Juditsky, 1992), AdamW (Loshchilov & Hutter, 2019), Nadam (Dozat), and

Adamax (Kingma & Ba, 2017)) and training with learning rate schedulers (i.e., StepLR and CosineAnnealingLR in PyTorch). The results of these experiments are shown in Table 3, where we retained only the best results for training with a scheduler, specifically those obtained using SGDM. IOMT consistently achieved the highest test accuracy among all the methods evaluated.

Table 4: Test accuracy (%) of the full training with more baselines.

|  | ASGD | AdamW | Ndam | Adamax | SGDM | stepLR | cosLR | ours |
|---|---|---|---|---|---|---|---|---|
| usps | 95.83 | 95.62 | 95.32 | 96.06 | 95.83 | 91.31 | 95.80 | **96.81** |
| mnist | 99.25 | 98.97 | 99.18 | 99.32 | 99.47 | 94.72 | 99.46 | **99.51** |

Additionally, we also conducted experiments using a complex imbalanced dataset ImageNet-A (Hendrycks et al., 2021), with the results displayed in Table 5. IOMT's dynamic adaptation to critical saddle points enhances performance on complex problems, resulting in over a 2% improvement in top-1 accuracy.

Table 5: Test accuracy (%) of the full training on ImageNet-A dataset.

|  | SGD | SGDM | Adagrad | RMSprop | Adam | SWATS | Padam | AdaBound | ours |
|---|---|---|---|---|---|---|---|---|---|
| acc@1 | 15.24 | 16.19 | 5.20 | 4.13 | 3.53 | 3.68 | 16.31 | 4.33 | **18.47** |
| acc@3 | 30.74 | 31.20 | 13.12 | 10.42 | 9.87 | 8.94 | 31.76 | 11.23 | **33.83** |
| acc@5 | 38.29 | 39.68 | 17.49 | 17.52 | 13.83 | 12.55 | 40.01 | 15.46 | **41.32** |

In addition to the results presented in Table 2, we also conducted more PEFT experiments on different models. Table 6 shows the experimental results. IOMT achieves superior test accuracy across more models and tasks.

Table 6: The testAcc. (%) and tuning time (sec.) for the vanilla method and our IOMT under the head fine-tuning. The time of IOMT includes the tuning time and the transferability estimation time.

| model | task method | usps testAcc. (%) | usps time (sec.) | mnist testAcc. (%) | mnist time (sec.) | stl10 testAcc. (%) | stl10 time (sec.) |
|---|---|---|---|---|---|---|---|
| ResNet18 | SGDM | $68.86_{\pm 0.71}$ | 534 | $71.59_{\pm 1.88}$ | 4443 | $91.87_{\pm 1.27}$ | 18551 |
| | Adam | $66.04_{\pm 0.11}$ | 543 | $71.16_{\pm 1.46}$ | 4876 | $91.02_{\pm 1.87}$ | 14909 |
| | SWATS | $68.55_{\pm 0.51}$ | 453 | $73.27_{\pm 0.38}$ | 4651 | $92.02_{\pm 1.86}$ | 14285 |
| | Padam | $65.32_{\pm 1.20}$ | 505 | $70.41_{\pm 0.52}$ | 4743 | $92.36_{\pm 0.36}$ | 19919 |
| | AdaBound | $67.80_{\pm 0.27}$ | 568 | $72.66_{\pm 0.31}$ | 4944 | $92.23_{\pm 1.37}$ | 34858 |
| | ours | $\mathbf{70.64}_{\pm 0.80}$ | 527+4 | $\mathbf{74.79}_{\pm 3.38}$ | 4801+6 | $\mathbf{93.56}_{\pm 0.41}$ | 13240+25 |
| ResNet152 | SGDM | $72.27_{\pm 0.12}$ | 4443 | $77.54_{\pm 1.02}$ | 7437 | $96.66_{\pm 0.32}$ | 78232 |
| | Adam | $71.73_{\pm 1.26}$ | 4876 | $77.64_{\pm 1.72}$ | 7636 | $96.11_{\pm 0.32}$ | 48541 |
| | SWATS | $71.82_{\pm 0.27}$ | 5121 | $78.47_{\pm 0.18}$ | 9211 | $96.66_{\pm 0.51}$ | 115560 |
| | Padam | $72.31_{\pm 0.69}$ | 5594 | $76.18_{\pm 1.17}$ | 8878 | $96.51_{\pm 0.39}$ | 107856 |
| | AdaBound | $72.94_{\pm 1.33}$ | 4508 | $78.33_{\pm 0.67}$ | 12934 | $95.44_{\pm 2.47}$ | 152592 |
| | ours | $\mathbf{74.36}_{\pm 0.41}$ | 5001+6 | $\mathbf{79.60}_{\pm 0.60}$ | 7637+8 | $\mathbf{97.60}_{\pm 0.28}$ | 44381+68 |
| MobileNet v2 | SGDM | $91.28_{\pm 0.39}$ | 18551 | $93.74_{\pm 0.77}$ | 50294 | $92.14_{\pm 0.71}$ | 11464 |
| | Adam | $89.86_{\pm 0.22}$ | 12237 | $93.82_{\pm 0.11}$ | 55024 | $91.31_{\pm 0.71}$ | 15208 |
| | SWATS | $90.68_{\pm 0.25}$ | 17921 | $93.90_{\pm 0.28}$ | 57655 | $92.77_{\pm 0.27}$ | 12971 |
| | Padam | $90.69_{\pm 0.67}$ | 18940 | $92.71_{\pm 0.22}$ | 51656 | $91.69_{\pm 0.29}$ | 12941 |
| | AdaBound | $91.36_{\pm 0.37}$ | 17328 | $93.67_{\pm 0.94}$ | 46955 | $91.39_{\pm 0.45}$ | 24367 |
| | ours | $\mathbf{92.32}_{\pm 0.39}$ | 13240+8 | $\mathbf{94.65}_{\pm 0.06}$ | 56121+11 | $\mathbf{93.28}_{\pm 0.79}$ | 14972+70 |

## D  DETAILS OF INDEPENDENT EXPERIMENTS

**The selection of optimizer switch strategy.** Compared to switching optimizers with simple strategies, IOMT employs the transferability assessment $\omega_t$ and variance halving in its acquisition function. To estimate the effectiveness of our design, we compared IOMT with two simple strategies

(random switch "random" and periodic replacement "cyclical") and two ablation versions (without transferability assessment "w/o $\omega_t$" and without variance halving "w/o halve"). The experimental result in Table 7 shows that simple random or periodic switching fails to produce high test accuracy. In addition, the usage of transferability assessment and variance halving both effectively enhance the adaptability of the current task, resulting in improved accuracy (up to 2%) and lower variance. Additionally, we also compared IOMT with these strategies under other models (i.e., ResNet152 "RN152" and MobileNet V2 "MN"), as shown in Figure 8 and Figure 9

Table 7: Experimental results for different switch strategies under full training on ResNet18.

| method | usps | | | mnist | | |
|---|---|---|---|---|---|---|
| | trainAcc. (%) | testAcc. (%) | time (sec.) | trainAcc. (%) | testAcc. (%) | time (sec.) |
| random | $98.36_{\pm 2.22}$ | $94.27_{\pm 1.41}$ | 225 | $98.91_{\pm 0.39}$ | $98.33_{\pm 0.46}$ | 1916 |
| cyclical | $97.38_{\pm 1.63}$ | $92.88_{\pm 1.94}$ | 223 | $99.49_{\pm 0.20}$ | $98.94_{\pm 0.25}$ | 1914 |
| w/o $\omega_t$ | $99.55_{\pm 0.27}$ | $95.82_{\pm 0.42}$ | 205 | $99.72_{\pm 0.09}$ | $99.25_{\pm 0.07}$ | 2101 |
| w/o halve | $99.54_{\pm 0.22}$ | $95.70_{\pm 0.35}$ | 226 | $99.67_{\pm 0.21}$ | $99.19_{\pm 0.08}$ | 2095 |
| ours | $\mathbf{99.67_{\pm 0.19}}$ | $\mathbf{96.51_{\pm 0.08}}$ | 227 | $\mathbf{99.85_{\pm 0.17}}$ | $\mathbf{99.48_{\pm 0.03}}$ | 1965 |

Table 8: The independent experimental result for transferability assessment.

| | metric | w/o $\omega_t$ | with $\omega_t$ |
|---|---|---|---|
| RN152 | trainAcc. (%) | 80.47 | **81.10** |
| | testAcc. (%) | 72.6 | **75.29** |
| | time (sec.) | 3531 | 3347+9 |
| MN | trainAcc. (%) | 98.38 | **98.42** |
| | testAcc. (%) | 91.57 | **91.93** |
| | time (sec.) | 26084 | 23271+8 |

Table 9: The independent experimental result for different optimizer switching strategies.

| | metric | random | cyclical | ours |
|---|---|---|---|---|
| RN152 | trainAcc. (%) | 76.56 | 77.79 | **81.10** |
| | testAcc. (%) | 71.49 | 71.95 | **75.29** |
| | time (sec.) | 3271 | 3299 | 3347+9 |
| MN | trainAcc. (%) | 97.58 | 98.01 | **98.42** |
| | testAcc. (%) | 90.33 | 90.73 | **91.93** |
| | time (sec.) | 24997 | 23712 | 23271+8 |

**The initial selection method.** We compared the impact of using random selection and weighted selection during the initial phase on subsequent training. In the experiments, we set the initial phase to 20 epochs and used a switch step size of $\tau = 25$.

**The model compression technique.** We compared the impact of using other compression techniques within IOMT on the results, as shown in Table 10.

**The optimizer space.** We examined how different candidate hyperparameter spaces affected the performance of IOMT. In addition to the original HP space described in Section 5.1 (where learning rate$\in$ [0.1, 0.01, 0.001]), we systematically expanded this space by including the following components: (1) weight decay for SGD, (2) momentum for SGDM, (3) weight decay for Adagrad, (4) weight decay for RMSprop, (5) alpha for RMSprop, and (6) weight decay for Adam. The ranges for these additional hyperparameters are as follows: weight decay values $\in$ [1e-2, 1e-3, 1e-4], momentum $\in$ [0.5, 0.6, 0.7, 0.8, 0.9], and alpha $\in$ [0.5, 0.6, 0.7, 0.8, 0.9]. Figure 11 illustrates the performance of the baseline methods compared to IOMT as the search space diversifies. For the baseline

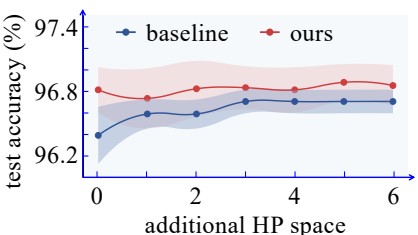

Figure 11: Test accuracy (%) for IOMT and baselines across various optimizer space on *usps*.

methods, we report the best accuracy achieved within the search space. Overall, IOMT demonstrates robust performance and consistently surpasses the baseline methods, even as the number of candidate hyperparameters increases.

Table 10: Experimental results for different compression techqniues under full training on ResNet18.

| method | usps | | | mnist | | |
|---|---|---|---|---|---|---|
| | trainAcc. (%) | testAcc. (%) | time (sec.) | trainAcc. (%) | testAcc. (%) | time (sec.) |
| RP | $\mathbf{99.72}_{\pm 0.12}$ | $\mathbf{96.44}_{\pm 0.17}$ | 201 | $99.61_{\pm 0.15}$ | $99.22_{\pm 0.18}$ | 1869 |
| UMAP | $98.56_{\pm 1.45}$ | $95.15_{\pm 2.15}$ | 253 | $99.73_{\pm 0.06}$ | $98.98_{\pm 0.13}$ | 2182 |
| PCA | $99.58_{\pm 0.13}$ | $96.34_{\pm 0.25}$ | 222 | $\mathbf{99.86}_{\pm 0.13}$ | $\mathbf{99.48}_{\pm 0.06}$ | 1974 |

**The hyperparameter settings.** In the experiments, we set the default hyperparameter configuration to $n_{ini} = 20$, $\tau = 25$, and n_components= 2. Based on the results presented in Figure 10, we have made the following observations.

- init step $n_{ini}$: Thanks to the ongoing updates of the surrogate model during training in IOMT, even a small initial step (i.e., $n_{ini} = 20$) can produce models with high test accuracy.

- switch step size $\tau$: A smaller step size facilitates quicker switching of the optimizer, which enhances accuracy (e.g., $\tau = 20\%$ of an epoch). Conversely, a larger step size makes it more challenging to collect training data for the surrogate model, resulting in a longer switching cycle and greater variance in the results.

- PCA components number: Selecting a small number of PCA components (for example, 2) can often yield good performance in IOMT. On the other hand, using a larger number of components may impair the surrogate model's ability to learn effectively, resulting in greater variance in test accuracy.

