# OpenReview forum: "Interleaving Optimizers for DNN Training"
_ICLR.cc/2025/Conference — Submitted to ICLR 2025_

### Official Review · Reviewer_8okb · 2024-10-30

**Soundness:** 2
**Presentation:** 2
**Contribution:** 2
**Rating:** 6
**Confidence:** 4

**Summary:**

The paper introduces the Interleaving Optimizer for Model Training (IOMT), a method for dynamically switching optimizers during a DNN's training to enhance model quality and convergence speed.  IOMT leverages surrogate models to estimate the performance of various optimizers based on parameter states and combines this with a transferability assessment to predict optimization gains. Furthermore, an acquisition function selects the optimizer with the highest expected benefit for each training stage. Experimental results suggest 3% accuracy increase on the CIFAR10 dataset and a 10% reduction in convergence time on STL10, positioning IOMT as a more efficient alternative to traditional or hybrid approaches.

**Strengths:**

The paper demonstrates a degree of coherence throughout, effectively guiding the reader through its arguments and findings. The choice of topic—optimizer selection—addresses a compelling area in machine learning research, and the approach presented brings a refreshing level of novelty to the field. Additionally, the paper’s examination of transferability represents a valuable addition, hinting at potential applications across various domains and suggesting broader implications for adaptability in optimization.
The comparative analysis presented in the paper appears thorough and fair, particularly with regard to benchmarking against both static optimizers and hybrid methods. However, as indicated in the weaknesses and questions section, some aspects of the methodology and analysis could benefit from further clarification or refinement to strengthen the overall impact of the work.

**Weaknesses:**

The paper presents two central claims that form the foundation of its proposed approach: first, that optimizers lead to distinct final weights by making divergent choices at key saddle points, and thus, selecting the most suitable optimizer at each of these saddle points could lead to more effective optimization of deep neural networks (DNNs). Secondly, the authors claim that the performance of each optimizer can be anticipated through a surrogate model. If this surrogate model is accurate, an acquisition function can be employed to switch between optimizers based on their predicted performance.

The design decisions surrounding the accuracy of the surrogate model, the effectiveness of the acquisition function, and the tuning of implementation hyperparameters are crucial elements in this framework. Each of these components directly influences the method's success, and so examining them carefully through the lens of methodological implementation, presentation, and results is warranted.

I will separate my critiques into the following themes:

### Method implementation:

First, the authors have not provided sufficient details on how they chose the hyperparameters for their method. For example, the initial phase, which relies on weighted random selection, is likely essential for achieving a high-quality surrogate model. This initial phase could be particularly sensitive to various hyperparameter configurations, impacting the model's effectiveness. Therefore, presenting the surrogate model's fit across different training points would help validate these design choices and clarify their impact.

Second, the switch step size is another significant hyperparameter that merits careful explanation, as it could substantially influence the optimization process. A more detailed rationale on how these hyperparameters were chosen, possibly accompanied by an ablation study or sensitivity analysis, would strengthen the methodological transparency and robustness of the proposed approach.

Finally, variance halving is presented as a core component of the method, yet no ablation study is included to demonstrate its necessity. Including this analysis would substantiate the importance of variance halving and justify its role within the overall framework.

### Presentation:

The presentation of the approach, while promising, could benefit from some improvements in clarity.

First, Figure 1 does not effectively communicate the paper’s key insight, as it displays only a single function evaluation and simply illustrates how different optimizers follow distinct paths from a saddle point. Replacing this with an aggregated figure that visualizes multiple runs would provide a clearer, more comprehensive view of how optimizers differ in their decision-making at saddle points.

Second, the explanation in the introduction could be clarified; specifically, the authors should more explicitly state that different optimizers result in varied weight configurations due to their unique decision-making characteristics. Establishing the significance of matching optimizers to saddle points would benefit the reader's understanding of the approach’s core motivation (Lines 40–48).

Third, Figure 7 is somewhat unclear. While it illustrates that different optimizers reach different final parameter states, it remains ambiguous whether these states yield comparable performance levels or achieve similar final accuracies. More detailed labeling or annotations could clarify these distinctions.

Finally, there is some ambiguity in Lines 14–19, where it is unclear whether the authors are jointly optimizing for both optimizers and other hyperparameters, or if their focus is solely on optimizer selection. Clarifying this would prevent potential confusion and ensure a clearer presentation of the research focus.

### Results:

The comparison with static and hybrid baselines is indeed informative; however, the observed incremental improvements on CIFAR may reflect the plateaued progress often seen on this dataset with current optimization methods. Given that the approach’s strength lies in its dynamic adaptability at critical saddle points, it would be beneficial to consider adversarial benchmarks like Imagenet-A to illustrate this advantage. Imagenet-A presents a challenging environment where subtle variations in optimization strategies could have a more pronounced impact, thereby highlighting the adaptability of the dynamic optimizer selection process.

Furthermore, the sensitivity of switching behavior to the surrogate model’s accuracy is an essential aspect. Since hyperparameters influence the surrogate model’s quality directly, understanding how these hyperparameters affect switching decisions would reinforce the robustness of the approach. One way to evaluate this would be to compare switching profiles under different surrogate strategies, including a scenario where the model operates suboptimally by relying purely on exploitation. Such an analysis would provide valuable insights into the flexibility of the proposed method, showing whether it can maintain performance even when the surrogate model is imperfect.

While optimizer selection is certainly an interesting approach and previous work, particularly Im et al., motivates the need for this, it still needs justification in terms of usability. Generally, tuning just the hyperparameters is simple and easy, which is why it is the go-to strategy for optimizing neural networks. A technique such as an optimizer selection, if necessary for the considered problems, should be able to beat a simple and common HPO baseline, such as tuning Learning rates with cosine annealing. Therefore, having this as a baseline can help empirically justify the need for optimizer selection.

**Questions:**

* How is this work related to learned optimization?
* How is the proposed SMBO technique related to previous work on Dynamic Algorithm Configuration (see Adriansen et al. JAIR’22)? As I see it, the aim here is to dynamically switch between optimization methods and that is understandable.
* How did the authors decide hyperparameters $n_{\text{ini}}$ and $\tau$?
* How did the authors decide on PCA as a feature engineering method? Is there a particular benefit that PCA has over other methods, say UMAP, Random projection or shallow autoencoders?
* How do the authors select which layers to use as inputs to the surrogate model?

---

> ### Author Response · Authors · 2024-11-26
> **Responses to Reviewer 8okb (1/3)**
>
> Thank you for your valuable comments, which improved our work. We sincerely appreciate the time and effort you dedicated to reviewing our submission. In light of your advice, we have made the necessary improvements to our draft.
>
> -----
>
> # Method implementation:
>
> > **The details on hyperparameter settings.**
>
> - We have added ablation studies in Section 5.3 to further investigate the impact of various hyperparameter settings in IOMT.
>
> > **1. The influence of hyperparameter configurations for the initial weighted selection phase.**
>
> - We have conducted additional experiments in Section 5.3 to analyze the designs of the initial phase. First, as shown in Figure 10(a), our proposed weighted selection achieves greater stability in subsequent training compared to the random sampling used in the initial phase. In addition, the results of the ablation study shown in Figure 10(b) indicate that while a longer initial phase may result in some improvements in accuracy, the ongoing updates to the surrogate model can still yield reasonably good training outcomes even with a smaller $n_{ini}$.
>
> > **2. The influence of the switch step size.**
>
> - We have added an ablation study regarding the switch step size in Section 5.3. The experimental results shown in Figure 10(b) indicate that frequently switching optimizers (i.e., using a smaller switch step size) typically achieve better accuracy performance.
>
> > **3. The analysis on variance halving.**
>
> - We have added an ablation experiment on variance reduction in Section 5.3, specifically in the ``w/o halve'' entry of Table 7 in Appendix D. The experimental results show that variance reduction helps improve the final accuracy while lowering variance.
>
> -----
>
> # Presentation:
>
> > **The presentation of the approach, while promising, could benefit from some improvements in clarity.**
>
> - We have revised the paper based on your suggestions to enhance clarity, including new visualizations, modified text descriptions, and clearer expressions.
>
> > **1. Replace Figure 1 with an aggregated figure.**
>
> - We have updated Figure 1 according to your suggestions. Specifically, we have conducted multiple training sessions using various optimizers on four commonly analyzed optimization functions: Rosenbrock, Himmelblau, Griewank, and Ackley. Each optimizer was trained 200 times with different hyperparameters and random seeds. The results demonstrate that different optimizers behave differently at saddle points, even when various hyperparameter settings are applied.
>
> > **2. The explanation in the introduction could be clarified (Lines 40–48).**
>
> - Thank you for your suggestions. We have revised the first paragraph of Section 1 to highlight the statement that "different optimizers result in varied weight configurations due to their unique decision strategies".
>
> > **3. Figure 7 is somewhat unclear. More detailed labeling or annotations could clarify these distinctions.**
>
> - We have added new visual examples (i.e., Figure 7a) to help clarify Figure 7, demonstrating that IOMT can achieve optimization paths/model weights that differ from those obtained by a single optimizer. Furthermore, we have revised the corresponding text in Section 4.2 to improve clarity. Specifically, we have added a description of the model construction method in Figure 7a, which involves mixing different numbers of optimizers (i.e., altering the optimizer space of IOMT) and visualizing the differences in the final model parameters obtained after training for the same number of epochs (i.e., 100 epochs).
>
> > **4. The author needs to clarify whether the authors are jointly optimizing for both optimizers and other hyperparameters, or if their focus is solely on optimizer selection (Lines 14-19).**
>
> - We have revised the text to emphasize the primary focus of our work. In particular, our research centers on switching between different types of optimizers, rather than selecting hyperparameters for a single optimizer.
>
> -----

---

> ### Author Response · Authors · 2024-11-26
> **Responses to Reviewer 8okb (2/3)**
>
> # Results:
>
> > **1. It would be beneficial to consider adversarial benchmarks like Imagenet-A to illustrate this advantage.**
>
> - We have conducted experiments on ImageNet-A and found that IOMT's dynamic optimizer selection leads to an improvement in accuracy (i.e., a top-1 accuracy increase of 2\%), as shown in Table 5 (cf., Appendix C.2). Due to time constraints, we have not yet completed experiments combining more advanced techniques or applying them to state-of-the-art models, but we believe IOMT would yield similar improvements in those cases. We will continue our experiments and update the results in future versions.
>
> > **2. The hyperparameter influence on the surrogate model’s quality (i.e., switching decisions).**
>
> - We have added additional independent experiments in Section 5.3 to explore the effects of various hyperparameter settings to switching results. Figures 10(b-d) demonstrate how variations in three key hyperparameters (i.e., the initial step size, the switch step size, and the number of PCA components) affect the experimental results. Although adjustments to these hyperparameters result in changes in test accuracy, a simple pattern emerges: larger initial step sizes, smaller switch step sizes, and fewer PCA components generally improve outcomes. This aligns with the conclusions presented in Section 5.1, which noted that the default hyperparameter settings yielded superior accuracy.
>
> > **3. The comparison with just tuning hyperparameter for a single optimizer, i.e., the simple HPO baselines like tuning Learning rates with cosine annealing.**
>
> - We have included a baseline for hyperparameter optimization (e.g., tuning the learning rate with cosine annealing), with the results presented in Table 4 (cf., Appendix D). In addition, We have added a discussion in Section 4.2 on the advantages of mixing different types of optimizers, i.e., ``IOMT not only integrates the optimization strategies of different optimizers but may also yield a searched model that a single optimizer cannot achieve''.
>
> -----

---

> ### Author Response · Authors · 2024-11-26
> **Responses to Reviewer 8okb (3/3)**
>
> # Questions:
>
> > **1. The relationship to learned optimization.**
>
> - "Learned optimization" can be viewed as a type of research in meta-learning (i.e., learning to learn). In this context, our work can be considered a form of learned optimization. However, learned optimizers are typically composed of black-box function approximators (such as neural networks) or adaptive hyperparameters tuners that are learned, as discussed in [1-3]. In contrast, our work focuses on learning to select among multiple categories of optimizers rather than learning a new black-box function or hyperparameters tuner. We have added relevant content in the related works.
>
> [1] Andrychowicz, Marcin, et al. "Learning to learn by gradient descent by gradient descent." *Advances in neural information processing systems* 29 (2016).
>
> [2] Daniel, Christian, Jonathan Taylor, and Sebastian Nowozin. "Learning step size controllers for robust neural network training." *Proceedings of the AAAI Conference on Artificial Intelligence*. Vol. 30. No. 1. 2016.
>
> [3] Harrison, James, Luke Metz, and Jascha Sohl-Dickstein. "A closer look at learned optimization: Stability, robustness, and inductive biases." *Advances in Neural Information Processing Systems* 35 (2022): 3758-3773.
>
> > **2. The relationship between SMBO technique and previous work on dynamic algorithm configuration (see Adriansen et al. JAIR’22).**
>
> - Thank you for the reference you provided. The problem we address falls under a type of "dynamic algorithm configuration", specifically referred to as "fine-grain tuning" in our draft. We have added relevant explanations in Section 3.2 to help better understand the content and significance of our research problem. Additionally, as pointed out in the reference, while the term "dynamic algorithm configuration" has been recently introduced, it encompasses a very broad range (cf. the first paragraph in Section 2.3). Previous work has explored "dynamic algorithm configuration" in contexts such as recursive algorithm selection, genetic algorithms, and learning rate control, whereas our focus is on the selection of optimizers. Additionally, while vanilla SMBO studies static algorithm configuration, IOMT applies this concept to dynamic algorithm selection.
>
> > **3. The selection of hyperparameters $n_{ini}$ and $\tau$?**
>
> - In the experiments, we chose $n_{ini} = 50$ and $\tau = 25$ as the default settings for IOMT (cf. Section 5.1) to obtain sufficient training data for the surrogate model and enough switching instances across different datasets. We have added ablation experiments to analyze the impact of hyperparameter settings on the results to demonstrate the selected default setting for IOMT.
>
> > **4. The choice of feature engineering method.**
>
> - We selected PCA as our feature engineering technique in IOMT because it is fast, effective, and easy to use. To further analyze the selection, we have conducted comparative experiments with other feature engineering methods (i.e., UMAP and random projection), and the results are presented in Table 10 (cf., Appendix D). We find that the simpler methods of PCA and random projection achieved higher test accuracy compared to the more complex UMAP method in the selected tasks (i.e., usps and mnist).
>
> > **5. The selection of layers for surrogate model’s input.**
>
> - IOMT selects parameters from specific layers rather than from all layers to reduce computational costs (cf. the revised Paragraph 3 on Section 3.3). For partial fine-tuning, IOMT focuses exclusively on the trainable weights, such as the matrices A and B in LoRA. For full training, IOMT includes the classifier layer along with some sampled hidden layers as inputs. In our experiments, we found that selecting only the classifier layer already yields good results.
>
> -----

---

> > ### Comment · Reviewer_8okb · 2024-11-26
> >
> > Many thanks for the thorough reply. The paper has improved quite a bit and is lot more self-contained. I increased my rating accordingly.
> >
> > However, I still have two concerns for which I would like to have your opinion:
> >
> > 1. Figure 10 shows that the tuning of the hyperparameters is important and the landscapes are not very consistent. For example, the switch step size looks a lot different between mnist and usps. This indicates that HPO is, in fact, needed for a more complex system.
> > 2. You argue in Section 4.2 that the overhead induced by your method is small. However, I wonder how small this overhead is compared to running a traditional optimizer, e.g., cosRL, for a bit longer. The differences in Table 4 are not significant and I simply wonder whether these could easily compensated.

---

> > > ### Author Response · Authors · 2024-11-27
> > > **Responses to Reviewer 8okb**
> > >
> > > We appreciate your acknowledgment of our revisions. Below are our responses to the concerns you raised.
> > >
> > > -----
> > >
> > > > **1. Hyperparameter tuning is important and landscapes are not very consistent.**
> > >
> > > Thank you for your reminder. We have included analyses to more effectively demonstrate the impact of hyperparameters (see the last paragraph in Section 5.3). The differences in the switch step size landscapes between the two datasets are primarily due to their sizes; MNIST contains approximately ten times more samples than USPS. We adjusted the x-axis to reflect epoch percentage (i.e., $\tau = x\% \times \text{epoch}$) and found that the landscapes show consistent behavior at this point (refer to the updated Figure 10c).
> > >
> > > -----
> > >
> > > > **2.1. How small is the overhead induced by IOMT compared to running a traditional optimizer.**
> > >
> > > The overhead is only 1.4% (smaller on large datasets and complex models). We have conducted an experiment with 500 iterations on the USPS dataset to analyze the overhead time associated with IOMT. The training time results for each method are presented below. It is evident that the overhead introduced by IOMT is relatively small and less than the time increase caused by cosLR. Furthermore, as the model size and dataset complexity increase, the advantages of our method in terms of overhead proportion and convergence speed become more pronounced (see Table 2 and Table 6).
> > >
> > > | #epoch | SGDM                                                         | SGDM (cosLR)                                                 | Adam                                                         | Adam (cosLR)                                                 | ours                                                         |
> > > | ------ |:------------------------------------------------------------:|:------------------------------------------------------------:|:------------------------------------------------------------:|:------------------------------------------------------------:|:------------------------------------------------------------:|
> > > | 100    | 203 $\pm$ 13 | 217 $\pm$ 34 | 207 $\pm$ 20 | 209 $\pm$ 20 | 206 $\pm$ 41 |
> > > | 500    | 1015 $\pm$ 171 | 1041 $\pm$ 129 | 987 $\pm$ 222 | 998 $\pm$ 205 | 992 $\pm$ 139 |
> > >
> > > -----
> > >
> > > > **2.2. Whether the differences in Table 4 could easily compensated.**
> > >
> > > The differences in Table 4 are not easy to surpass because the experimental results of the baselines in Table 4 are obtained through hyperparameter tuning (with lr = [0.1, 0.01, 0.001]) and 5 different random seeds. Additionally, we applied the cosLR method across various optimizers (including Adam, Adagrad, and SGDM), but the best result from SGDM still falls short of IOMT (cf. Page 16, Paragraph 1). The specific results and variance information are shown in the table below.
> > >
> > > |       | SGDM+cosLR                                                   | Adagrad+cosLR                                                | RMSprop+cosLR                                                | Adam+cosLR                                                   | ours                                                         |
> > > |:-----:| :------------------------------------------------------------: | :------------------------------------------------------------: | :------------------------------------------------------------: | :------------------------------------------------------------|:------------------------------------------------------------:|
> > > | usps  | 95.80 $\pm$ 0.67 | 93.67 $\pm$ 2.20 | 94.02 $\pm$ 2.29 | 95.28 $\pm$ 0.13 | 96.81 $\pm$ 0.21 |
> > > | mnist | 99.46 $\pm$ 0.06 | 99.44 $\pm$ 0.08 | 99.21 $\pm$ 0.10 | 99.19 $\pm$ 0.03 | 99.51 $\pm$ 0.01 |
> > >
> > > In contrast, the results for IOMT are obtained using only the default settings ($n_{ini}=50$ and $\tau=25$). Furthermore, as analyzed in Figure 11 regarding the increase in the hyperparameter search space, a more detailed search for the baselines does not lead to a test accuracy that surpasses IOMT. This is also because IOMT can achieve optimization paths/model weights that differ from those obtained by a single optimizer, even with varying hyperparameter settings.
> > >
> > > -----

---

### Official Review · Reviewer_1WPP · 2024-11-01

**Soundness:** 2
**Presentation:** 3
**Contribution:** 3
**Rating:** 6
**Confidence:** 3

**Summary:**

The paper introduces the IOMT framework, which combines multiple optimization algorithms in a novel way to enhance the training of deep learning models. This approach aims to leverage the strengths of different optimizers at various stages of training. The proposed approach builds a surrogate model that takes the hyper-parameters and the preprocessed model parameters as input and returns the here-defined score. An optimizer, hyper-parameter values, and the training time are then selected based on the acquisition function defined using the trained surrogate model. The authors conduct extensive experiments to validate the effectiveness of the IOMT approach, demonstrating its advantages over baseline (single) optimization methods and other hybrid methods. The results show improved performance in various scenarios with different models and different datasets.

**Strengths:**

A novel method for optimizing deep learning training by interleaving multiple optimizers. This innovative strategy allows for leveraging the unique strengths of different optimizers at various stages of training, which can lead to improved model performance.

Empirical validation. The paper provides extensive experimental results that demonstrate the effectiveness of the IOMT approach.

**Weaknesses:**

The problem is formulated as a general problem in (1). However, in the experiments of this paper, only a single hyper-parameter (i.e., learning rate) is considered and the training time is fixed. It is not clear whether it is effective when more hyper-parameters are to be optimized and the training time is to be optimized. In particular, it is not clear if the proposed approach can really select the best approach when baseline approaches have different number of hyper-parameters, leading to different difficulties in building surrogate models for different approaches.

Related to the above point, the search space is small. Only 5 methods with 3 possible learning rate values, resulting in 15 choices in total. It is not clear how much it scales.

Some details are not clear. How are w_p and w_d defined? It is recommended that their definitions be added to the Appendix.

**Questions:**

How could the proposed approach avoid overfitting? Doesn't it overfit if you run it longer? I couldn't find any mechanism that could prevent it.

---

> ### Author Response · Authors · 2024-11-26
> **Responses to Reviewer 1WPP**
>
> We sincerely appreciate your dedicated time and effort in reviewing our submission. We have carefully revised the manuscript according to your useful comments. Below is our response to your suggestion.
>
> -----
>
> > **W1&W2: Concerns on generalization to more hyper-parameters and search space.**
>
> A1: We have added experiments for IOMT and baselines for IOMT and baselines with an expanded hyperparameter space (cf. Paragraph 5 on Section 5.3 and the penultimate paragraph on Page 17). Under a broader range of hyperparameters, IOMT still achieves better test accuracy than baselines. Additionally, IOMT demonstrates stable performance even in an expanded search space, as shown in Figure 11.
>
> -----
>
> > **W3: The definitions of w_p and w_d can be added to the Appendix. (Sec. 3.4)**
>
> A2: We have added the definitions of $\omega_p$ and $\omega_d$ in Appendix A. In the 3rd paragraph on Page 6, we provide that $\omega_p$ refers to performance-based metrics (i.e., accuracy), while $\omega _d$ pertains to distribution-based metrics, including LogME and Leep.
>
> -----
>
> > **Q1: Potential overfitting of the proposed approach.**
>
> A3: The variance reduction described in Equation 7 on Page 6 can help alleviate overfitting. IOMT tends to select more stable optimizers (i.e., lower consideration for variance) at the late stage of training, which helps prevent overfitting to noise in the training data (similar to how a learning rate scheduler operates). Meanwhile, commonly used techniques for preventing overfitting—such as early stopping based on the validation set and learning rate schedulers—can also be applied to IOMT.
>
> -----

---

> > ### Comment · Reviewer_1WPP · 2024-11-26
> >
> > Thank you for the additional experiments. I have questions regarding the additional experiment:
> >
> > 1. In Figure 11, the authors changed the number of HPs. But in the text (L901-912), only one additional HP is described for each algorithm. What are the other HPs?
> >
> > 2. From Figure 11, I can see that the proposed approach is robust against the additional HPs in this experiment. However, because no other approach is tested on the same search space, the superiority of the proposed approach over the baseline approaches is not revealed. Don't the baseline approaches exhibit stable performance for varying number of HPs?

---

> ### Author Response · Authors · 2024-11-27
> **Responses to Reviewer 1WPP**
>
> Thank you again for your comments to our work. Below are our responses to the additional experiments.
>
> -----
>
> > **1. The changed HPs in Figure 11.**
>
> We have revised the corresponding descriptions and Figure 11 provide a clearer explanation of the additional hyperparameters. Specifically, we incrementally added the following hyperparameter search spaces in the experiments: (1) SGD weight decay, (2) SGDM momentum, (3) Adagrad weight decay, (4) RMSprop weight decay, (5) RMSprop alpha, and (6) Adam weight decay, in order to explore the impact of increasing the search space on the baselines and IOMT.
>
> -----
>
> > **2. The performance of baseline approaches to varying numbers of HPs.**
>
> We have updated Figure 11 to include the performance of the baseline approaches (i.e., the results with the highest accuracy within the total search space). It can be observed that IOMT consistently outperforms the baseline test accuracy as the search space increases.
>
> -----

---

> ### Comment · Reviewer_1WPP · 2024-11-27
>
> Thanks again for the clarification. Now I see the usefulness of the proposed approach, but I still have two concerns below.
>
> 1. The search space dimension is increased by 6, but only one hyperparameter is added for each optimizer, meaning that only two hyperparameters are considered for each algorithm. Therefore, the search space is still rather limited.
>
> 2. The baseline approach also performs robustly, implying that the added hyper-parameters are not quite difficult to optimize.

---

> > ### Author Response · Authors · 2024-11-28
> > **Responses to Reviewer 1WPP**
> >
> > We appreciate your assistance and acknowledgment of our improvements. Below are our responses to the two new concerns you raised.
> >
> > -----
> >
> > > **The search space is still rather limited.**
> >
> > We selected the commonly tuned HPs in the experiments of Figure 11. Moreover, in our experiments, we have found that IOMT yields stable results on other HPs too, consistent with the conclusions in Figure 11. We will organize the results and add them to the draft. Please note that the hyperparameter variations among different optimizers (e.g., conditional HPs that some HPs only exist in one type of optimizer) introduce extra complexity in the experimental result presentation, so we did not include them in the early version, but we will add them to the draft for better supporting our findings.
> >
> > -----
> >
> > > **The added hyperparameters are not quite difficult to optimize in the baselines.**
> >
> > In fact, the hyperparameter optimization in the baselines is difficult. The baseline results we present are based on the best outcomes from exhausted searches, which take significantly more time than IOMT. The configurations that yield the best results for the baselines under different seeds are completely different, with various optimizers achieving top-1 performance. In comparison, the training time for IOMT is approximately equal to that of a single training run using one optimizer with one set of HPs.
> >
> > -----
> >
> > As Thanksgiving approaches, we want to express our gratitude for your comments to our work which help improve our work. Wishing you a joyful Thanksgiving!

---

### Official Review · Reviewer_xWct · 2024-11-04

**Soundness:** 3
**Presentation:** 2
**Contribution:** 2
**Rating:** 5
**Confidence:** 4

**Summary:**

Deep neural networks often face the issue of how to select optimizers. This paper addresses this problem by proposing IOMT, a fine optimizer switch based on a surrogate method (i.e., Gaussian process). The authors provide relevant experiments to support their method. However, the experimental section has several issues, such as limited improvement in performance and a lack of ablation studies.

**Strengths:**

1.	The paper is easy to read.
2.	The problem of optimizers is clear.

**Weaknesses:**

1.	There exist some typos/grammatical errors in the paper and should be revised.
2.	The authors mention in the abstract that the main issue with DNNs is uncertain impact of optimizers. However, they have not rigorously validated or proven this issue on larger datasets (ImageNet, COCO2017, VOC) and tasks (i.e., object detection, VQA, NLP-based tasks). Clearly, this statement needs further verification to establish its existence.
3.	In page one, what is the definition of better model quality?
4.	In page 1, the author claims that surrogate models, namely Gaussian process will predict the performance of different optimizers during training, the presentation of performance is unclear: is it accuracy or something else?
5.	Why transferability assessment is used to predict training cost? I don’t see any definition and discussion in your paper.
6.	The presentation of Figure 1 is not clear.
7.	Please provide more experiments on training cost.
8.	Please provide a comparison with more competitors.
9.	This topic is not novel. Please read the paper about AutoML.
10.	It would be interesting to use NAS to search for the best configuration for optimizers.
11.	Provide a more thorough discussion of the generalization ability, robustness, and potential applications of the proposed approach.
12.	The format of references is not correct.
13.	The main limitation of this paper is that proposed method lacks theriacal analysis, ablation study, larger datasets (ImageNet, COCO2017, VOC), more tasks (i.e., object detection, VQA, NLP-based tasks), and larger networks (i.e., stable diffusion, LLMs), and visualization.
14.	Exploring the reasons behind the success of these techniques and providing intuitive explanations would contribute to the overall scientific contribution of the work.

**Questions:**

see Weaknesses

---

> ### Author Response · Authors · 2024-11-26
> **Responses to Reviewer xWct**
>
> Thanks for your useful suggestions. We have carefully revised our draft based on your comments and addressed each point in detail below.
>
> ------
>
> > **W1: The typos/grammatical errors.**
>
> A1: We have carefully revised the manuscript and corrected typographical and grammatical errors.
>
> ------
>
> > **W2 & W7: Further verification on more datasets and tasks.**
>
> A2: We have conducted experiments on a more complex dataset (i.e., ImageNet-A), and included additional baselines (e.g., AdamW and CosineAnnealingLR). On the ImageNet-A dataset, IOMT achieves significantly higher accuracy than other methods, reaching 2% in top-1 accuracy (cf., Table 5 on Appendix C.2). Meanwhile, we have conducted additional ablation studies in Section 5.3. Due to time constraints, we will include more experiments on additional tasks and datasets in future versions.
>
> ------
>
> > **W3: The definition of better model quality in page 1.**
>
> A3: The term "better model quality" in the draft refers to the improved performance results of the trained model, such as higher accuracy. We have added corresponding explanations in the text to aid understanding in Paragraph 1 of Page 2.
>
> ------
>
> > **W4: The presentation of performance predicted by surrogate models is unclear: is it accuracy or something else?**
>
> A4: The term "performance" refers to the loss reduction achieved by various optimizers, as shown in Equation 2. We have revised the corresponding description in Section 1 for better clarity.
>
> ------
>
> > **W5: The reason for using transferability assessment.**
>
> A5: We have refined the relevant statements in the 3rd paragraph on Page 6 to highlight the motivation of using transferability assessment. Specifically, the transferability between pre-trained models and the current task reflects the correlation between upstream and downstream tasks, which affects the ease of fine-tuning. For example, transferring a model pre-trained on the MNIST dataset to the SVHN dataset is easy because both datasets involve handwritten digits. In contrast, transferring between MNIST and CIFAR-10 is more challenging, as they have different characteristics.
>
> ------
>
> > **W6: The presentation of Figure 1 is not clear.**
>
> A6: We have updated Figure 1 and added annotations for clarity.
>
> ------
>
> > **W8: Comparison with more competitors.**
>
> A7: We have incorporated additional baselines (e.g., AdamW, NAdam, and CosineAnnealingLR). The experimental results are detailed in Table 4 (cf., Appendix C.2).
>
> -----
>
> > **W9: This topic is not novel. Please read the paper about AutoML.**
>
> A8: We have added a corresponding explanation in the discussion of Section 4.1 to highlight the novelty of our work compared to current AutoML methods. Although IOMT is closely related to hyperparameter optimization, a key topic in AutoML, traditional methods generally use globally consistent static tuning or adaptive learning rate tuning, whereas we focus on fine-grain tuning of the optimizer during the training process.
>
> -----
>
> > **W10: Use NAS to search for the best configuration for optimizers.**
>
> A9: We have added a hyperparameter tuning strategy (i.e., automatic learning rate tuning) and broadened the hyperparameter search space for searching the best configuration. The experimental results are in Table 4 and Figure 11 on Appendix. Compared to baselines with more optimally searched configurations, IOMT provides more robust outcomes and achieves superior test accuracy.
>
> -----
>
> > **W11: More discussion of the generalization ability, robustness, and potential applications.**
>
> A10: We have added further discussion about IOMT’s generalization ability and potential applications through a new visual analysis (Figure 7a in Section 4.2). Additionally, we have conducted ablation studies to examine the impact of various hyperparameters and search spaces on the robustness of IOMT (cf. Section 5.3).
>
> -----
>
> > **W12: The format of references is not correct.**
>
> A11: We have modified the citations in the draft which using the \citet{} format according to the ``Formatting Instructions for ICLR 2025 Conference Submissions''. We would like to know if the reviewer has identified any additional formatting issues for the references.
>
> -----
>
> > **W13: Lacks theoretical analysis, ablation study, larger datasets, more tasks, larger networks, and visualization.**
>
> A12: We have added more experiments (cf. as in A2). Additionally, we have included new visual analyses in the revised draft (e.g., Figure 1 and Figure 7).
>
> -----
>
> > **W14: Exploring the reasons behind the success of these techniques and providing intuitive explanations would contribute to the overall scientific contribution of the work.**
>
> A14: We have included a visual analysis in Section 4.2 that highlights the advantages of IOMT. This analysis demonstrates how IOMT can identify optimization paths that a single optimizer cannot achieve. Additionally, we have conducted new experiments in Section 5 to further investigate the rationale behind the design of IOMT.
>
> -----

---

> > ### Comment · Reviewer_xWct · 2024-11-26
> >
> > Thanks for your reply.  My concerns have not been fully resolved. Therefore, I cannot judge whether the current version meets the standard for ICLR publication. I will keep my score.

---

> > > ### Author Response · Authors · 2024-11-27
> > > **Responses to Reviewer xWct**
> > >
> > > Thanks for your feedback. Could you please let us know your concerns that have not been fully resolved? We are more than happy to further clarify and resolve them.

---

### Official Review · Reviewer_xcs3 · 2024-11-09

**Soundness:** 3
**Presentation:** 2
**Contribution:** 3
**Rating:** 6
**Confidence:** 3

**Summary:**

This paper proposes an optimizer switching method, the interleaving optimizer for model training (IOMT), for DNN training. The proposed IOMT automatically switches the optimizer and hyperparameters during the model training based on the optimization status information, resulting in faster convergence and better model quality. The experimental result demonstrates that IOMT outperforms existing SGD optimizers on several tasks and models.

**Strengths:**

- The motivation for switching the optimizer during the model training is good.
- The switching strategy based on the Gaussian process surrogate seems novel, and some modified acquisition function is introduced.
- The effectiveness of the proposed method is demonstrated by several datasets and DNN models.

**Weaknesses:**

- The reason for the improvement of model quality is not sufficiently elaborated.
- It is unclear how sensitive IOMT is against its hyperparameter setting. Therefore, it would be great if the sensitivity analysis of the hyperparameter setting in IOMT was conducted.
- The reviewer feels that there are several unclear points of the detailed procedure of IOMT. Please see questions.

**Questions:**

- The reviewer does not understand why IOMT improves the model quality, although the faster convergence by IOMT is somewhat convincing. Could you explain why IOMT can improve the generalization performance of the trained model?
- The reviewer is not sure what weights parameter sets are used for computing principle axes and when the PCA calculation is performed. To compute the principle axes (eigen decomposition), data points $\{ {\bf \theta} \}$ should be prepared.
- IOMT introduces additional hyperparameters, such as the number of PCA components and the switch step size $\tau$. What is the impact of such hyperparameter settings on performance?
- Most optimizers have internal states, such as momentum information. How are such internal states inherited and maintained when switching the optimizer?
- There are some typos: "Iterleaving" should be "Interleaving."



----- After authors' response -----

Thank you for your response. I am satisfied with the authors' response. I would keep my score positive side.

I would like to comment a little bit more about the PCA calculation.
Let us assume the dimensionality of $\theta$ parameters as $D$; then, ideally, we need more than $D$ data points of $\theta$ parameters to calculate the principal axes. At least, I suppose that we need $D'$ data points for PCA calculation, i.e., {$\theta^{(1)}, \dots, \theta^{(D')}$ }, $\theta^{(k)} \in \mathbb{R}^D$, where $D'$ is the number of PCA components for dimensionally reduction.
I am still not sure how the authors collect such $\theta$ parameter sets.

---

> ### Author Response · Authors · 2024-11-26
> **Responses to Reviewer xcs3**
>
> Thanks for your useful suggestions. We have thoroughly revised our submission based on your comments. Below is our response to your suggestion.
>
> ------
>
> > **W1 & Q1: The reason for IOMT’s performance improvement. (Sec. 4.2)**
>
> A1: Switching optimizers in IOMT can help find the optimization paths that a single optimizer cannot achieve (cf. Section 4.2). This expanded range of search paths results in improvements in model quality (i.e., higher accuracy). To clarify this, we have included new visualization examples in Section 4.2 Figure 7(a) to illustrate this situation better.
>
> ------
>
> > **W2 & Q3: The impact and sensitivity of hyperparameter settings for IOMT.**
>
> A2: We have added ablation experiments for the hyperparameter settings (cf. Section 5.3). Figure 10(b-d) presents the experimental results, showing that variations in hyperparameters affect the performance of IOMT according to certain rules. Below is a summary of our findings.
>
> - init step $n_{ini}$: Thanks to the continuous updating of the surrogate model during training in IOMT, even a small init step (i.e., $n_{ini}=20$) can obtain models with high accuracy. While a larger initial step, which provides more initial experience, may lead to a slight yet inconsistent improvement in test accuracy.
>
> - switch step size $\tau$: A smaller step size facilitates quicker switching of the optimizer, which enhances accuracy in small datasets (e.g., \textit{usps}). Conversely, a larger step size makes it more challenging to collect training data for the surrogate model, resulting in a longer switching cycle and greater variance in the results.
>
> - PCA components number:  Choosing a small number of PCA components (e.g., number=2) can already provide good performance in IOMT. In contrast, using a larger number of components may hinder the surrogate model's ability to learn effectively, leading to increased variance in test accuracy.
>
> ------
>
> > **W3 & Q2: Unclear about the input parameters and time for PCA calculation.**
>
> A3: IOMT selects the key layers as input for PCA calculation, such as the trainable layers in partial fine-tuning and the classifier layer for full training (cf. Paragraph 2 on Page 5). We have revised the relevant content in Section 3.3 to clarify these points.
> Furthermore, PCA calculations are conducted at the start of each optimizer switch cycle (i.e., every $\tau$ iterations). We have updated Section 3.1 and Figure 3 for clarity.
>
> ------
>
> > **Q4: The internal states inheritance when switching the optimizer.**
>
> A4: IOMT does not choose to retain internal states when switching optimizers. Our method emphasizes the "short-term" benefits that each optimizer can achieve given the current parameter state (cf. Section 3.3 Paragraph 4). Each time switching optimizers, IOMT treats that transition as the start of a new optimization process.
>
> ------
>
> > **Q5: typos: "Iterleaving" -> "Interleaving."**
>
> A5: We have fixed it.
>
> ------

---

### Author Response · Authors · 2024-11-26
**Global Response**

Dear Area Chair, SPC, and Reviewers,

We sincerely appreciate your dedicated time and effort in reviewing our submission. The valuable comments have helped us improve our submission. We have revised the manuscript and are now submitting the revised pdf along with this summary of the revisions. To facilitate differentiation, we have highlighted key modifications in the updated pdf in blue. The main changes in the revised submission are as follows.

- **Experiments:** we have conducted several ablation studies to further analyze the IOMT design further. These studies include: (i) initial sample methods, (ii) model compression techniques,  (iii) considerations of variance and training process in the acquisition function, (iv) variations in optimizer search space size, and (v) the setting of IOMT's hyperparameters (i.e., the initial steps $n_{ini}$, the switch step size $\tau$, and the number of PCA components). Furthermore, we have carried out additional experiments to assess the performance of our method across more baselines (e.g., NAdam, AdamW, CosineAnnealingLR) and datasets (e.g., Imagenet-A).

- **Methodology:** we have added new visual examples and experimental analyses to show the advantages of IOMT and to explore the reasons behind its higher accuracy.

- **Presentation:** we have corrected typographical errors and enhanced the overall presentation with clearer figures and more concise wording.


Sincerely,

Authors of submission \#9286

---

### Meta-Review · Area_Chair_xS6x · 2024-12-22

**Metareview:**

This paper introduces the Interleaving Optimizer for Model Training (IOMT), a framework for dynamically switching optimizers during neural network training. It leverages surrogate models to estimate optimizer performance and selects among them with an acquisition function. Reviewers agreed on the novelty of the approach, clear problem definition, and empirical validation through extensive experiments. They criticized a lack of clarity in some methodological details and of experiments on larger datasets and tasks. Most important to me would be the aspect of experiments on larger datasets and tasks. I encourage the authors to benchmark it on AlgoPerf to compare to properly tuned baselines -- if possible that would be a a strong demonstration of the method's empirical strength.

**Additional Comments On Reviewer Discussion:**

The authors addressed some concerns by conducting additional experiments and improving explanations. Despite these improvements, concerns about scalability and robustness persisted for some reviewers.

---

### Decision · Program_Chairs · 2025-01-22

Reject